# Nonequilibrium self-assembly dynamics of icosahedral viral capsids packaging genome or polyelectrolyte

Maelenn Chevreuil[1,2], Didier Law-Hine[1], Jingzhi Chen[1], Stéphane Bressanelli [2], Sophie Combet[3], Doru Constantin[1], Jéril Degrouard[1], Johannes Möller[4], Mehdi Zeghal[1] & Guillaume Tresset [1]

The survival of viruses partly relies on their ability to self-assemble inside host cells. Although coarse-grained simulations have identified different pathways leading to assembled virions from their components, experimental evidence is severely lacking. Here, we use time-resolved small-angle X-ray scattering to uncover the nonequilibrium self-assembly dynamics of icosahedral viral capsids packaging their full RNA genome. We reveal the formation of amorphous complexes via an en masse pathway and their relaxation into virions via a synchronous pathway. The binding energy of capsid subunits on the genome is moderate (~7$k_B T_0$, with $k_B$ the Boltzmann constant and $T_0 = 298$ K, the room temperature), while the energy barrier separating the complexes and the virions is high (~20$k_B T_0$). A synthetic polyelectrolyte can lower this barrier so that filled capsids are formed in conditions where virions cannot build up. We propose a representation of the dynamics on a free energy landscape.

[1] Laboratoire de Physique des Solides, CNRS, Univ. Paris-Sud, Université Paris-Saclay, 91405 Orsay Cedex, France. [2] Institute for Integrative Biology of the Cell (I2BC), CEA, CNRS, Univ. Paris-Sud, Université Paris-Saclay, 91198 Gif-sur-Yvette Cedex, France. [3] Laboratoire Léon Brillouin (LLB), UMR 12 CEA-CNRS, Université Paris-Saclay, CEA-Saclay, 91191 Gif-sur-Yvette Cedex, France. [4] European Synchrotron Radiation Facility (ESRF), 71 avenue des Martyrs, 38000 Grenoble, France. Correspondence and requests for materials should be addressed to G.T. (email: guillaume.tresset@u-psud.fr)

The simplest icosahedral viruses can be viewed as nanometer-scaled protein shells called capsids encasing the genome in the form of nucleic acids. The genome is packaged during replication in the host cell[1] through an efficient self-assembly process[2–4]. The fact that some viruses are able to self-assemble from purified components and to package materials in vitro has promoted the development of engineered viral capsids enclosing synthetic polymers[5,6], heterologous nucleic acids[7–10], nanoemulsion droplets[11], gold[12,13], and magnetic[14] nanoparticles to name a few. Therefore, understanding the molecular mechanisms of viral self-assembly can help devise therapeutic strategies for inhibiting viral replication[15,16] and design functional biocompatible nanocontainers[17] for drug delivery and medical imaging. Furthermore, viruses can also help address fundamental questions on the physical principles underlying multicomponent self-assembly, notably by elucidating the dynamic pathways leading to error-free complex structures[18,19].

Capsid subunits are subject to a delicate balance between a short-range, mostly hydrophobic, attraction and a long-range electrostatic repulsion, while the subunit–genome interaction is electrostatically attractive[2,20,21]. Molecular dynamics simulations[22–24] have shed light on two pathways that depend on the interaction energies between components[25,26]: (i) in the first pathway, a piece of capsid is formed on the genome then grows by sequential binding of capsid subunits. Subunit binding and genome packaging occur in a synchronous manner. This pathway takes place mainly for strong subunit–subunit attractive interactions and is often called nucleation growth or nucleation elongation in the field; (ii) in the other pathway, subunits adsorb onto the genome en masse in a short time frame and the resulting amorphous complex relaxes into an ordered, filled capsid. This pathway is preferentially followed for strong subunit–genome attractive interactions.

The self-assembly and disassembly dynamics of empty icosahedral viral capsids has been investigated by time-course static light scattering[27–29], resistive-pulse sensing through nanofluidic channel[30], atomic force microscopy[31,32], and time-resolved small-angle X-ray scattering (TR-SAXS)[33–36]. In the presence of genome, by contrast, data are scarce because the dynamics is intrinsically more complex and the coexistence of different components renders the interpretation difficult. Kler et al.[37] observed by TR-SAXS the synchronous pathway during the packaging of short RNA segments in SV40-derived nanoparticles. However, because SV40 is a DNA virus and its genome is actually 10-fold larger than the segments used in that study, the observations may not be relevant in a biological context. Single molecule fluorescence correlation microscopy[38,39] suggested the en masse pathway for the genome packaging in satellite tobacco necrosis virus capsid, but the technique provides the hydrodynamic radius as a function of time, and thereby lacks information on the morphology and on the mass of species.

The cowpea chlorotic mottle virus (CCMV) is an icosahedral, single-stranded (ss)RNA plant virus widely used in physical[6,8,28] and nanotechnological[9,10,19,40] studies. Its $T = 3$ capsid is made up of 90 chemically identical protein dimers—hereafter called subunits—enclosing the genome shared between four ssRNA segments. In the absence of genome, the subunits remain dissociated at neutral pH, but spontaneously form empty capsids at acidic pH due to globally enhanced attractive interactions[41,42]. With genome, recent investigations[8,43] have shown that, while subunits barely interact with it at neutral pH and high ionic strength, amorphous nucleoprotein complexes (NPCs) are formed when the ionic strength is lowered. In turn, at acidic pH, these NPCs relax into well-ordered, icosahedral virions. Roughly speaking, the ionic strength controls the subunit–genome interactions and the pH acts on the subunit–subunit interactions.

In this study, we conduct a quantitative investigation on the nonequilibrium self-assembly dynamics of CCMV capsids packaging their full genetic material. Using TR-SAXS with a high brilliance synchrotron source, we establish the existence of the en masse and synchronous pathways by tuning the subunit–subunit and subunit–genome interactions, and we estimate several related nonequilibrium quantities. We also investigate the role played by the cargo (genome or polyelectrolyte) in the pathway selection.

## Results

**NPCs self-assemble through an en masse pathway.** At pH 7.5 and an ionic strength of 0.5 M, CCMV subunits are in the form of dissociated dimers[42] with a radius of gyration $R_g$ of 33 Å (Supplementary Fig. 1). CCMV genome consists of four ssRNA segments distributed in such a way that 2980 nucleotides on average are packaged per virion. Therefore, for the sake of simplicity, we hereafter refer to ssRNA molecules of this reference size as the genome. Static scattering measurements on purified genome gave a $R_g$ of 140 ± 11 (s.e.m.) Å (Supplementary Fig. 2).

The subunits have two positively charged, flexible, N-terminal arms (Fig. 1a, inset) that allow them to bind on the genome by electrostatic interactions. We performed static measurements by small-angle neutron scattering (SANS) with a mixture of subunits and genome in 68% $D_2O$ at pD 7.5. In this fraction of heavy water, the contrast between RNA and the solvent vanishes, and solely the scattering intensity arising from the proteins is detected[6]. The subunit-to-genome mass ratio $\rho$ in a native virion is about 3.6, but we performed experiments with $\rho$ around 6, which has been found to be the minimal ratio ensuring the complete packaging of genome[43]. Figure 1a shows that whereas the scattering intensity was near the noise level at an ionic strength of 0.5 M, it increased by one order of magnitude at 0.1 M, which can be ascribed to the binding of subunits on the genome, thus forming NPCs (see drawings in Fig. 1a). By assuming that the solution contained free subunits and $N$th-order complexes ($N = 0, 1, 2\ldots, +\infty$), each of which was made of $N$ subunits bound on the genome, we can infer an upper limit $\langle N \rangle_{up}$ of the mean value of $N$ from the forward scattering intensities $I_0 \equiv I(q = 0)$ (see Methods). Accordingly, we estimated that each NPC contained on average a maximum of $\langle N \rangle_{up} = 75 \pm 31$ (s.e.m.) subunits. The extraction of $\langle N \rangle_{up}$ from the scattering data does not require any prior knowledge on the distribution of $N$, and therefore, this independent-model quantity was used across this study to probe the binding of subunits on the genome in various conditions.

We carried out TR-SAXS experiments by rapidly mixing purified subunits and genome with a final ionic strength of 0.1 M. Figure 1b depicts the assembly of NPCs for $\rho = 6$. Subunits were bound on the genome very rapidly and their mean number per NPC remained stable for more than 10 min at about 77 subunits. A single exponential decay function indicated a binding time $\tau_{bind}$ of 28 ms. By contrast, the variation of $R_g$ occurred over a structural relaxation time $\tau_{struc}$ of 48 s, that is, three orders of magnitude higher than $\tau_{bind}$. We used a double exponential decay for fitting $R_g$: the first phase might reflect a compaction of the genome due to the presence of subunits, while the second phase might result from the partial association of bound subunits into capsid subassemblies. These findings suggested that NPCs were formed through an en masse pathway (see drawing of Fig. 1b): subunits were first captured by the genome in ~100 ms, then NPCs slowly self-organized over several minutes. Interestingly, other experiments performed in the same conditions showed that $\tau_{struc}$ was reduced at small values of $\rho$ (Supplementary Fig. 3): the

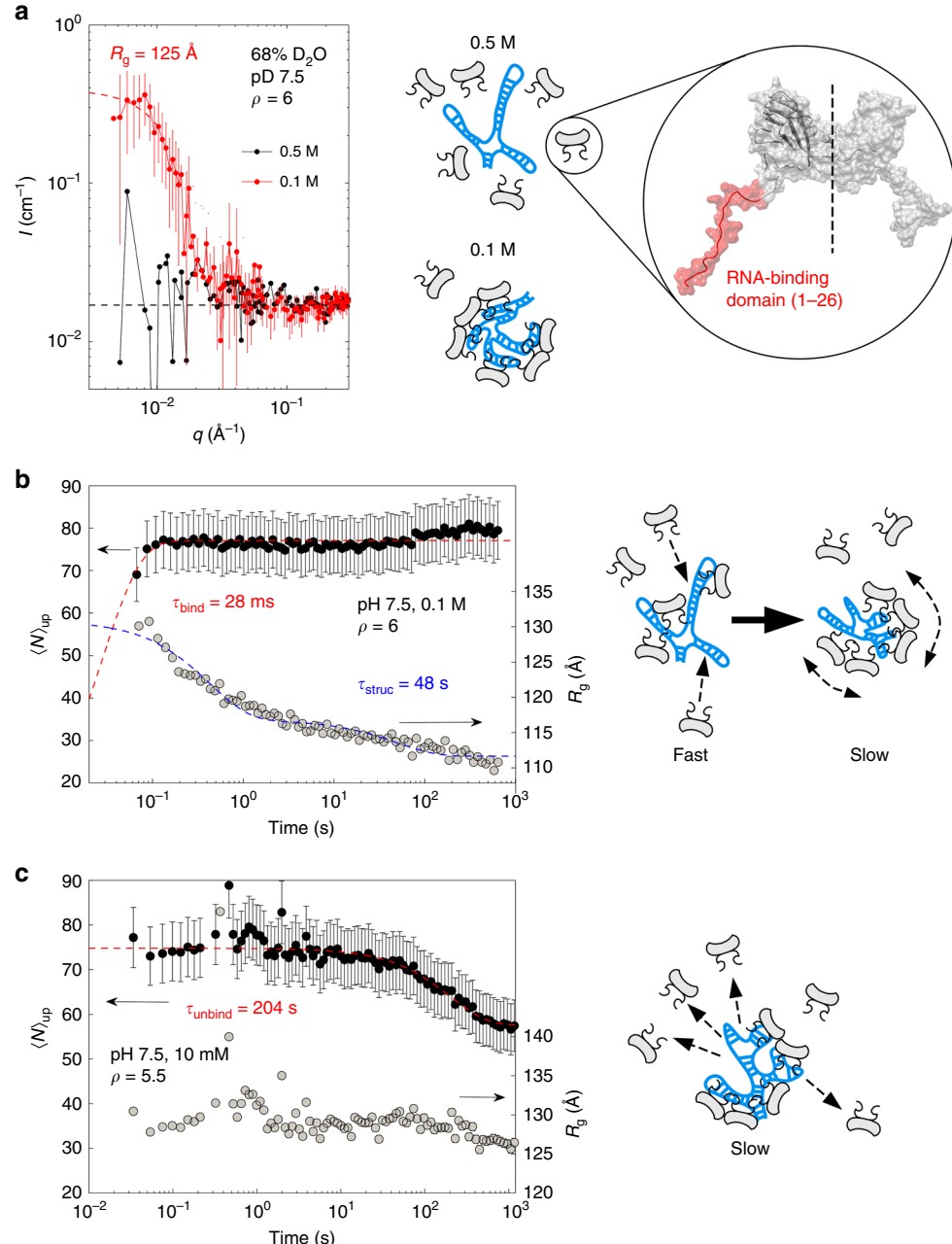

**Fig. 1** Self-assembly dynamics of NPCs. **a** SANS patterns of CCMV subunits and genome in 68% $D_2O$ at pD 7.5 and for ionic strengths of 0.5 M (black symbols) and 0.1 M (red symbols). The red dashed line is a Guinier approximation and the black dashed line indicates the noise level. The subunit-to-genome mass ratio $\rho$ was 6. The drawing depicts the subunits (in light gray) and the genome (in blue) at the two ionic strengths. (inset) Structure of a CCMV dimeric subunit. The flexible RNA-binding domain in red contains 26 residues, 10 of which carry one cationic charge[44]. **b** $\langle N \rangle_{up}$ (black discs) and radius of gyration $R_g$ (light gray discs) as a function of time after mixing subunits and genome at an ionic strength of 0.1 M for $\rho = 6$. The red and blue dashed lines are single and double exponential decay functions yielding the binding time $\tau_{bind}$ and the structural relaxation time $\tau_{struc}$, respectively. The red dashed line starts from the origin since, in this experiment, $\langle N \rangle_{up}(t=0) = 0$. The drawing illustrates the en masse pathway of NPC assembly. **c** $\langle N \rangle_{up}$ (black discs) and $R_g$ (light gray discs) as a function of time after a twofold dilution of NPCs pre-assembled at an ionic strength of 10 mM and $\rho = 5.5$. The red dashed line is a single exponential decay function giving the unbinding time $\tau_{unbind}$. The drawing depicts the release of subunits upon dilution. Error bars are all defined as s.e.m. and were obtained by propagating the standard deviations of neutron or photon counts (see Methods)

compaction of the genome, which is thought to correspond to the fast phase, may slow down when a large number of subunits bind on it as is the case at high values of $\rho$. The subsequent self-organization of capsid subassemblies, that is, the second phase, may be affected accordingly.

The experiment depicted on Fig. 1c shows a solution containing NPCs pre-assembled at an ionic strength of 10 mM and $\rho$ = 5.5, and rapidly diluted twofold. $\langle N \rangle_{up}$ decreased slowly from 75 to 58 subunits per NPC with an unbinding time of 204 s. The variation of $R_g$ was not significant over the period of observation (~1200 s). We think that it was due to two competing effects: the NPCs tended to be less compact upon the release of subunits, but at the same time, the released subunits did not contribute any longer to the radius of gyration of NPCs. $R_g$ remained therefore roughly unchanged. This experiment demonstrates that dilution is sufficient to release subunits from NPCs within a few minutes

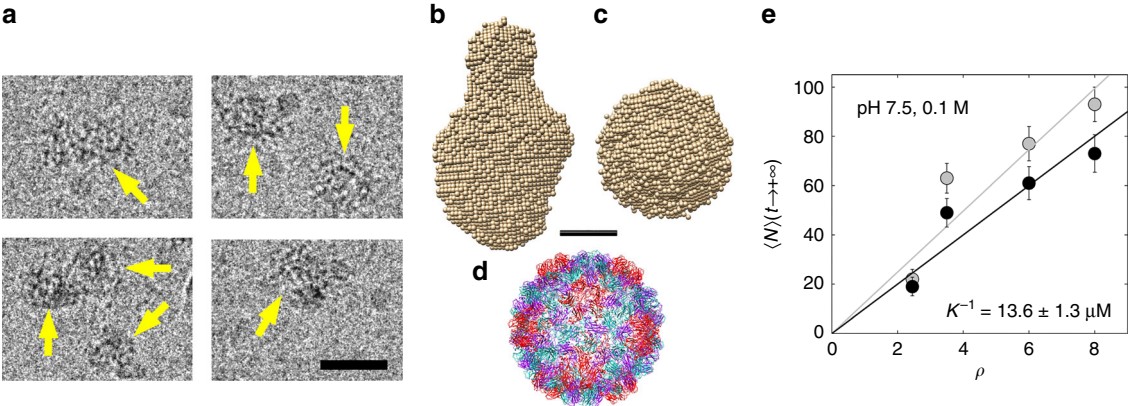

**Fig. 2** NPCs at equilibrium. **a** Cryo-TEM images of NPCs (yellow arrows) obtained in the same conditions as in Fig. 1b after a few days of incubation. Scale bar is 30 nm. **b** Side and **c**, top views of an ab initio reconstruction of NPCs calculated from the last scattering pattern of the experiment shown in Fig. 1b. **d** Crystal structure of the native capsid shown for comparison (PDB reference 1ZA7). Scale bar is 10 nm. **e** $\langle N \rangle$ obtained near equilibrium for various mass ratios and evaluated from the forward scattering intensities either by using the templated assembly model (black discs) or by computing $\langle N \rangle_{up}$ (light gray discs; see Supplementary Fig. 7 for the time traces). The black and light gray lines are linear fits of $\langle N \rangle$ obtained by the templated assembly model and $\langle N \rangle_{up}$, respectively. The critical concentration $K^{-1}$ is estimated from the slope of the black line. Error bars are defined as s.e.m. and were obtained by propagating the standard deviations of photon counts (see Methods)

and suggests a weak binding energy between the subunits and the genome.

The morphology of NPCs at equilibrium is shown in Fig. 2a by transmission electron cryomicroscopy (cryo-TEM): we can see amorphous species with an equivalent diameter of $27 \pm 4$ (s.d.) nm and an average aspect ratio of 1.13, that is, a size which is close to that of the native virion (~29 nm in diameter). We carried out an ab initio shape determination of NPCs with a model of densely packed beads from the scattering pattern collected at the end of the assembly experiment performed at $\rho = 6$. The low resolution structure appeared elongated with a length of ~380 Å and a cross-sectional diameter of ~230 Å (Fig. 2b–d), in fair agreement with the cryo-TEM images.

In order to get a quantitative insight into the binding thermodynamics and the diffusion of subunits during the NPC self-assembly, we analyzed the scattering intensities with a simple templated assembly model (see Supplementary Methods), that is,

$$C_N + S \underset{k^-}{\overset{k^+}{\rightleftarrows}} C_{N+1}, \tag{1}$$

where $S$ denotes the free subunits, $C_N$ are the $N$th-order complexes with $N = 0, 1, 2..., +\infty$ and $C_0$ stands for the bare genome. $k^+$ and $k^-$ are the forward and backward reaction rate constants, respectively. At equilibrium, the mean number of subunits per NPC is given by $\langle N \rangle (t \to +\infty) \approx (c_S - K^{-1})/c_G$, where $c_S$ and $c_G$ are the total molar concentrations of subunits and genome respectively, and $K^{-1} = k^-/k^+$ is the inverse of equilibrium constant, also equal to the critical subunit concentration above which NPCs are formed (Supplementary Fig. 4). For the sake of simplicity, we kept $k^+$ and $k^-$ constant, which implied that the effects of the interactions between subunits, whether they were bound on the genome or free, were averaged over $N$. Figure 2e (black discs) shows $\langle N \rangle$ estimated near equilibrium for experiments carried out at various $\rho$, by using an analytical expression relating it with $I_0(t \to +\infty)$ through the templated assembly model (see Supplementary Methods). The total subunit concentration $c_S$ was fixed in all the experiments, which implied that $\langle N \rangle (t \to +\infty) \propto (1 - K^{-1}/c_S)\rho$, and thus we estimated $K^{-1}$ from a linear fit (black line) to be $13.6 \pm 1.3$ (s.e.m.) μM. The fact that the experimental curve was linear indicated that the NPCs were still far from being saturated by bound subunits. At higher values of $\rho$, we expect the number of

bound subunits to reach a limit and $\langle N \rangle (t \to +\infty)$ will exhibit a plateau (Supplementary Fig. 5). It follows from $K^{-1}$ that the binding free energy between the subunits and NPCs was $G_{bind} \simeq -17$ kJ mol$^{-1} = -6.7k_B T_0$ (see Supporting Information for the expression of $G_{bind}$), with $k_B$ the Boltzmann constant and $T_0$ the room temperature (298 K). A linear fit of $\langle N \rangle_{up}$ as a function of $\rho$ (light gray line) returned a $K^{-1}$ of $10.9 \pm 1.8$ (s.e.m.) μM and $G_{bind} \simeq -6.9k_B T_0$, that is, a very similar value. This value is moderate and indicates that the subunits were indeed loosely bound to the NPCs. Recall, however, that $G_{bind}$ also comprises an effective contribution arising from the subunit–subunit interactions.

Furthermore, if the binding of subunits was diffusion-limited, the forward reaction rate constant entering the templated assembly model would be expressed by $k^+ = 4\pi N_A R_C D_S$, where $R_C$ is a typical radius for NPCs taken as 120 Å, $D_S$ is the apparent diffusion coefficient of subunits, and $N_A$ is the Avogadro constant. By fitting $I_0$ collected from an experiment at $\rho = 8$ with the intensity given by the templated assembly model (Supplementary Fig. 6), we estimated $D_S$ to be $4.9 \times 10^{-12}$ m$^2$ s$^{-1}$, while the diffusion coefficient of a single subunit computed by molecular dynamics simulations amounted to $3.0 \times 10^{-10}$ m$^2$ s$^{-1}$, that is, two orders of magnitude higher. This discrepancy indicates an energy barrier, possibly arising from the subunits in NPCs hindering the insertion of incoming free subunits through steric and electrostatic repulsions. This barrier likely also arises from an orientation dependence of the incoming subunits as well as from a dependence on the subunit conformation, notably at the level of the RNA-binding domain. At any rate, the binding process is partly reaction-limited.

**NPCs relax into virions via a synchronous pathway.** By lowering the pH of a solution of NPCs, the subunit–subunit interactions become more attractive. As a result, the NPCs become metastable and slowly relax into species structurally identical to native virions[8]. In the experiment depicted on Fig. 3a, we mixed a solution of pre-assembled NPCs with a buffer bringing the pH to 5.2 and the ionic strength to 55 mM. $\langle N \rangle_{up}$ increased from 75 to about 85 in more than 1 h, which gave a long binding time of 3100 s. The radius of gyration did not change significantly during this time frame (a variation of a few angstroms compared to 11 nm is below the typical resolution of small-angle scattering).

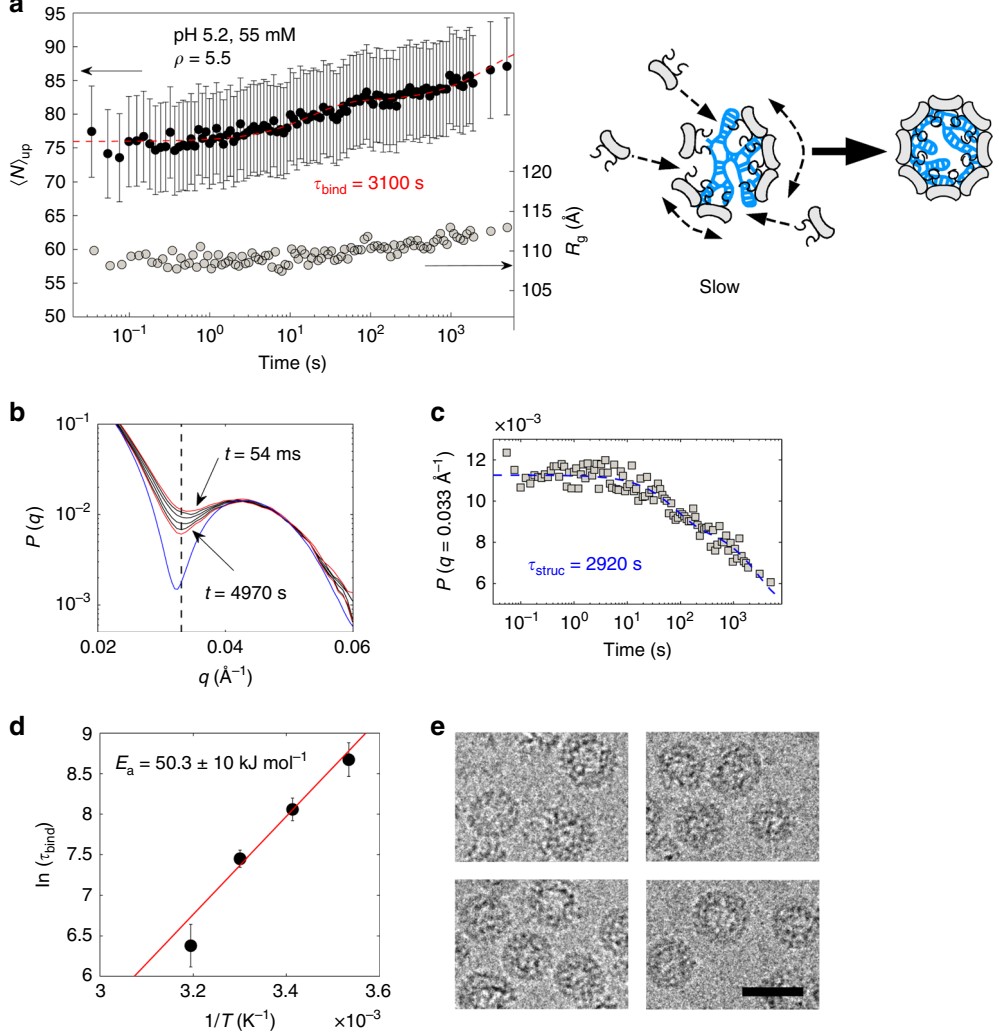

**Fig. 3** Relaxation dynamics of NPCs into virions. **a** $\langle N \rangle_{up}$ (black discs) and $R_g$ (light gray discs) vs. time after mixing pre-assembled NPCs with a buffer solution bringing the pH to 5.2 and the ionic strength to 55 mM. $\rho = 5.5$ and the temperature was 20 °C. The red dashed line is a double exponential decay function yielding the binding time $\tau_{bind}$ as the longest decay time. The drawing illustrates the synchronous pathway where NPCs are self-organizing while capturing free subunits in order to form virions. **b** Form factors $P(q)$ of NPCs (red and black solid lines) at different time points (see Supplementary Fig. 8 for methodological details). The red solid lines correspond to the first and last measurements collected at 54 ms (top curve) and 4970 s (bottom curve), respectively. The blue solid line is the form factor of purified virions and the black dashed line indicates $q = 0.033$ Å$^{-1}$. **c** Value of the form factors of NPCs at $q = 0.033$ Å$^{-1}$ vs. time. The blue dashed line is a double exponential decay fit yielding an estimate for the structural relaxation time $\tau_{struc}$. **d** Logarithm of $\tau_{bind}$ as a function of the inverse of temperature for different experiments performed at 10, 20, 30, and 40 °C (see Supplementary Fig. 10 for the corresponding traces of $\langle N \rangle_{up}$). The red line is a linear fit that gives the activation energy $E_a$. **e** Cryo-TEM images of reconstituted virions obtained in the same conditions as in (**a**) after a few days of incubation. Scale bar is 30 nm. Error bars are all defined as s.e.m. and were obtained by propagating the standard deviations of photon counts (see Methods)

However, a close inspection of the form factors of NPCs show the arches becoming more pronounced over time, indicating a structural self-organization (Fig. 3b). A comparison with the form factor of purified virions showed that the relaxing NPCs had a size and a structure close to those of native virions, and that empty capsids were not in detectable amounts (Supplementary Fig. 8). The evolution of the form factor at its first minimum reflects the dynamics of this self-organization process and we estimated the structural relaxation time $\tau_{struc}$ to be 2920 s (Fig. 3c). Note that we used again a double exponential decay function for the fittings: the first phase might be fast because several binding sites in the NPCs were readily accessible to the free subunits, while the second phase might take more time because the dynamics became limited by the slow construction of the capsid. Since $\tau_{bind} \approx \tau_{struc}$, the system likely proceeded

through a synchronous mechanism where the capsid was building up by capturing free subunits while packaging the genome. Simultaneously, a few bound subunits could continue being exchanged with the bulk solution or even transferred between NPCs through collisions. We carried out the same relaxation experiment at various temperatures ranging from 10 °C up to 40 °C. For an energetically activated process, the binding time should follow an Arrhenius-like law, that is, $\tau_{bind} \propto \exp(E_a / k_B T)$, where $E_a$ is the activation energy. Figure 3d plots the logarithm of $\tau_{bind}$ as a function of $1/T$ and shows a linear relationship consistent with the existence of an energy barrier $E_a$ estimated to be $20 k_B T_0$. We also investigated the shortest decay time of the binding process and we found out that the fast phase might be activated as well with an energy of $11 k_B T_0$ (see Supplementary Fig. 9a). At all the temperatures, $\tau_{struc}$ remained of the same order of magnitude

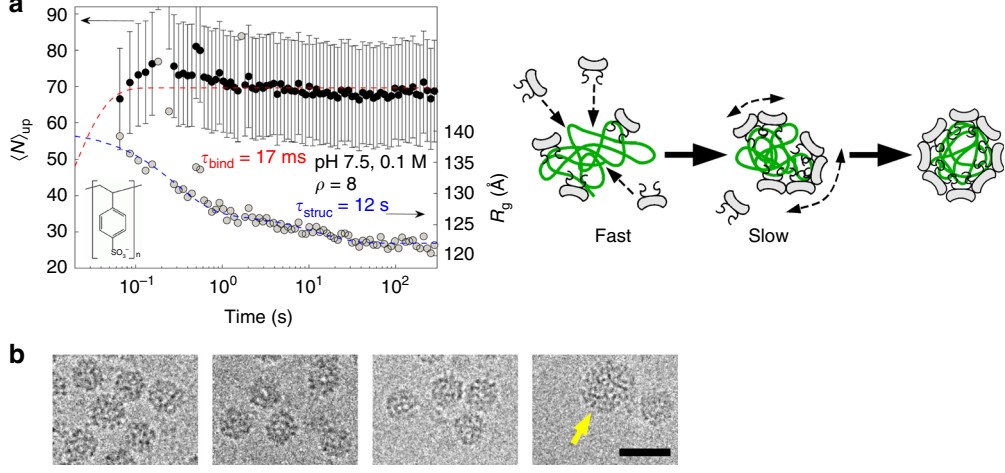

**Fig. 4** Self-assembly dynamics of PSS-filled capsids. **a** $\langle N \rangle_{up}$ (black discs) and $R_g$ (light gray discs) vs. time after mixing subunits and 600-kDa poly(styrene sulfonic acid) (PSS) at pH 7.5 with a final ionic strength of 0.1 M. The subunit-to-PSS mass ratio was $\rho = 8$. The red and blue dashed lines are single and double exponential decay functions yielding $\tau_{bind}$ and $\tau_{struc}$, respectively. The red dashed line starts from the origin since, in this experiment, $\langle N \rangle_{up}(t = 0) = 0$. Error bars are defined as s.e.m. and were obtained by propagating the standard deviations of photon counts (see Methods). The chemical structure represents the PSS repeat unit. The scattering patterns are shown on Supplementary Fig. 12. The drawing next to the graph depicts the en masse pathway with the subunits in light gray and the PSS in green. **b** Cryo-TEM images of PSS-loaded capsids obtained at $\rho = 6$ after a few days of incubation. The yellow arrow indicates a large species. Scale bar is 30 nm

as $\tau_{bind}$, including for the fast phase (Supplementary Figs. 9b, 10, 11). We checked out the morphology of the species by cryo-TEM after a few days of incubation (Fig. 3e). Like the native virions, they displayed a diameter of $29 \pm 1$ (s.d.) nm and a hollow core arising from the fact that the genome is packed against the interior surface of the capsid[45] due to electrostatic interactions.

**PSS-filled capsids are formed via an en masse pathway.** Poly (styrene sulfonic acid) (PSS) is a flexible negatively charged synthetic polyelectrolyte, whose linear charge density ($\sim 0.33 e^- \text{ Å}^{-1}$) is close to that of ssRNA. We used 600-kDa PSS which carried about 3100 negative charges on average, that is, an amount similar to that of the viral genome. However, subunits and PSS mixed at pH 7.5 produce spherical species[6]. Figure 4a shows the time evolution of $\langle N \rangle_{up}$ estimated by assuming an average of 1.5 PSS chains packaged per capsid, as previously established by SANS[6]. $\langle N \rangle_{up}$ plateaued at $\sim$70—which was consistent with the presence of objects smaller than native virions (90 subunits)—with a binding time $\tau_{bind}$ of 17 ms very close to the $\tau_{bind} = 28$ ms obtained with the genome in the same conditions. Likewise, the radius of gyration relaxed in $\tau_{struc} = 12$ s, which was of the same order of magnitude as the value of 48 s found with the genome. The time evolution of $R_g$ exhibited two phases, which were most likely of the same nature as those observed with the genome (Fig. 1b). As a consequence, since $\tau_{bind} \ll \tau_{struc}$, PSS-filled capsids self-assembled via an en masse pathway: the subunits were rapidly bound on the polyelectrolyte and the resulting complexes slowly self-organized into spherical species. Cryo-TEM images in Fig. 4b show that the final capsids were well assembled but slightly less ordered than the native virions, which suggested that the capsids comprised defects. The vast majority had a diameter of $23 \pm 1$ (s.d.) nm and only a few species were larger than 29 nm, in good agreement with our previous results[6]. The fact that capsids were able to close up in the presence of PSS but not with genome at pH 7.5 could be explained by the hydrophobicity and the flexibility of PSS. PSS is known to be insoluble in water when its charge fraction drops below 30% owing to its hydrophobic backbone[46]. We therefore think that the RNA-binding domains of the bound subunits might neutralize a large fraction of the negative charges carried by the PSS. The latter, which is very

flexible, might then collapse through a coil–globule transition that could allow the bound subunits to contact each other and to build up a closed shell, despite the energy barrier related to their electrostatic repulsion and to the conformational change required to arrange onto an icosahedral lattice. A recent theoretical study demonstrated that flexible, linear chains require less free energy than stiff, branched ones to be confined in a capsid[47], which is in qualitative agreement with our results. Like virions, PSS-filled capsids are stable against dilution, which indicates that, at the end of assembly, the subunits making up the capsid are no longer able to exchange with the bulk solution.

## Discussion
The present study shows experimentally how the self-assembly dynamics of icosahedral virions is influenced by the subunit–subunit and subunit–genome interactions. Garmann et al.[8,48] reported that during the assembly of CCMV NPCs performed with exogenous RNA derived from the genome of brome mosaic virus and of tobacco mosaic virus, all subunits were bound to RNA regardless of the subunit-to-RNA mass ratio well beyond the ratio of 3.6 found in native virions. This finding entails the irreversibility of the binding and, subsequently, a high binding energy of subunits on NPCs. In contrast, the binding energy was found here, with the native genome, to be moderate ($\sim 7 k_B T_0$) despite its electrostatic nature. This is more consistent with competition experiments where subunits initially bound to exogenous RNA eventually migrated toward viral RNA as long as the NPCs had not yet relaxed into virions[43]. Such a moderate binding energy may thus be advantageous for the virus, allowing it to select its genome specifically. It is also in agreement with our observation of 75 subunits per NPC at most in conditions where the subunit-to-genome molar ratio was 150 ($\rho = 6$) because a significant fraction of subunits remained free. Yet, these NPCs relaxed into well-formed virions, that is, comprising 90 subunits, upon lowering the pH. We emphasize that, at pH 7.5, the subunits have a negative net charge and the attractive force with the genome is ensured by the subunit dipole moment. Molecular dynamics simulations revealed that the dipole moment was reduced subsequently to the folding of the RNA-binding domains

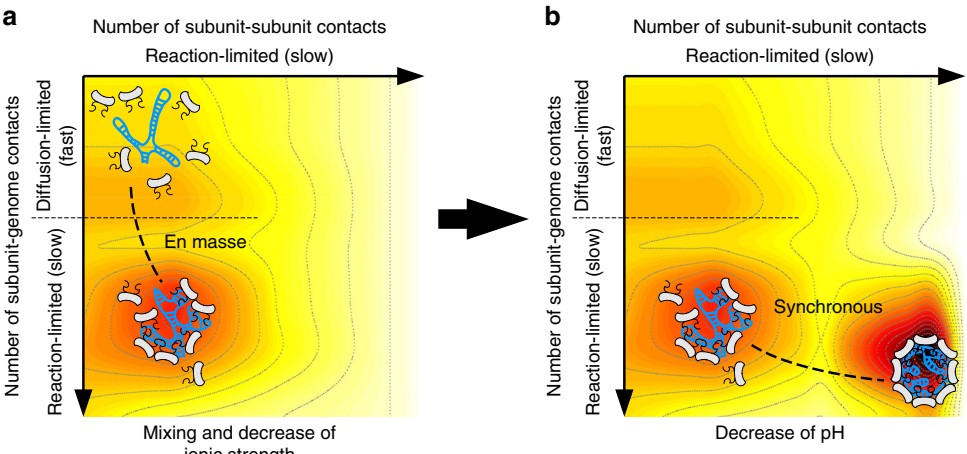

**Fig. 5** Schematical representations of the free energy landscape in varying ionic conditions. Dark colors stand for low free energies, while bright colors represent high free energies. **a** This graph illustrates the mixing of subunits with genome and a change of ionic strength at neutral pH, which leads to the formation of NPCs through an en masse pathway. **b** This graph illustrates the subsequent relaxation of metastable NPCs into virions upon a change of pH through a synchronous pathway

against the body of the proteins (Supplementary Fig. 13), thus contributing to reduced subunit–genome interactions.

In agreement with coarse-grained simulations[24], our measurements provided evidence that the en masse and synchronous pathways were selected according to the interactions tuned here by the pH and ionic strength. In host cells, cooperativity between subunits and genome may be required to package and therefore to protect the genome within a short time frame. A separate synchronous step in turn prevents misassembly—including exogeneous RNA packaging—and kinetic traps. In this respect, note that when we tried to perform a one-step assembly by mixing subunits and genome while simultaneously lowering the pH and the ionic strength, a lot of misassembled and aggregated objects were formed. What may drive the assembly in vivo is still unclear. A possible candidate was divalent cations $Ca^{2+}$ that are known to stabilize the capsid[44]. However, our experiments performed in the presence of $Ca^{2+}$ did not reveal any significant change in the self-assembly dynamics of NPCs (Supplementary Fig. 14). Surprisingly, increasing subunit–subunit interactions by acidification was not required for PSS-filled capsids, which demonstrates that, in certain conditions, the subunits are able to build up a closed shell in a single step at neutral pH. We hypothesize therefore that the assembly occurs in vivo at neutral pH and that divalent cations have little role to play at this stage. NPCs may be formed gradually as the subunits are expressed, which permits an efficient selectivity for the genome, and then, when the number of subunits per NPC reaches a high enough value—probably well above 100—the NPCs irreversibly relax into icosahedral virions. This scheme has the advantage that it provides a simple mechanism—namely, capsid protein expression—to switch to the assembly step of the viral cycle, a crucial regulation level for positive-sense (ss) RNA viruses such as CCMV whose RNA serve not only as genome but also as messenger RNA and replication templates.

Figure 5 summarizes the self-assembly dynamics on a free energy landscape parametrized by two reaction coordinates, that is, the number of subunit–subunit contacts (x-axis) and the number of subunit–genome contacts (y-axis). Along the x-axis, the dynamics is reaction-limited with a high energy barrier to overcome as encountered in the synchronous pathway. Along the y-axis, the dynamics is diffusion-limited only as long as only a few subunits are bound on the genome. Beyond a certain number of subunits, the dynamics becomes reaction-limited because more collisions of incoming subunits with NPCs are required for successful insertions. The free energy landscape can be tailored by adjusting the pH and ionic strength, but also depends on the flexibility and hydrophobicity of the polyelectrolyte to package.

Elucidating the dynamic pathways pertaining to viral replication will be useful to combat viral infection. But viruses are also a great source of inspiration for biomimetic nanomaterials, especially for synthetic delivery systems made of oppositely charged components such as polymer–DNA and lipid–DNA complexes used for gene delivery applications[49]. The atomic precision with which their building blocks form so complex a structure and their stability in a variety of adverse environments should foster their use outside their natural context. The present work will hopefully contribute to strengthening the efforts devoted to understanding, modeling, and harnessing the assembly of a wide range of viruses, viral nanocages, and synthetic delivery systems.

## Methods

**Sample preparation.** CCMV was grown in blackeye cowpea leaves (*Vigna unguiculata*), then capsid proteins and genomic RNA were extracted from purified virions[6,35,36]. More precisely, 140 g of infected leaves were mixed and stirred with 300 mL of 0.15 M sodium acetate, pH 4.8, and 300 mL of ice-cold chloroform prior to 10 min of centrifugation at $10,000 \times g$. The virions in the supernatant were precipitated by adding NaCl to a final concentration of 20 mM and 8% (v w$^{-1}$) of poly(ethylene glycol) (MW 8000). The solution was centrifuged at $10,000 \times g$ for 10 min. The pellet was resuspended and the solution centrifuged at $8000 \times g$ for 10 min. The supernatant was centrifuged through a 20% (v w$^{-1}$) sucrose cushion at $150,000 \times g$ for 2 h and the virions in the pellet were stored at $-80$ °C. For protein purification, virions were dialyzed against 50 mM Tris-HCl, pH 7.5, 0.5 M CaCl$_2$, 1 mM EDTA, pH 8, and 1 mM dithiothreitol (DTT). The solution was centrifuged at $150,000 \times g$ for 18 h and pure proteins collected in the supernatant were stored at 4 °C in 50 mM sodium acetate, pH 4.8, 0.5 M NaCl, and 1 mM EDTA. Viral RNA was purified by mixing 0.1 mL of virions with 1 mL of TRIzol® reagent (Life Technologies, France). The sample was centrifuged at $12,000 \times g$ for 15 min and the aqueous phase was removed prior to addition of 0.5 mL of isopropanol. After a 10-min centrifugation at $12,000 \times g$, the pellet was washed with 1 mL of 75% ethanol and centrifuged at $7500 \times g$ for 5 min. The RNA pellet was resuspended in RNase-free water and stored at $-80$ °C until use. Virions, capsid proteins, and genomic RNA were deemed pure for $A_{260}/A_{280} > 1.6$, $A_{280}/A_{260} > 1.65$, and $A_{260}/A_{280} > 1.8$, respectively.

Six hundred kilodaltons PSS was sulfonated[6] from polystyrene (Polymer Source, Canada). Briefly, 100 mg of polystyrene was dissolved in 15 mL of cyclohexane. Four microliters of a mixture of concentrated sulfuric acid and phosphoric acid was added and the solution was agitated for 2 h at 50 °C. Ten milliliters of ice water was added, and then the aqueous phase was filtered and dialyzed against pure sterile water for 2 weeks. PSS was freeze-dried and stored until use. Elemental analyses indicated an average degree of sulfonation of 95%.

All samples were dialyzed against their final buffer no more than 2 days before experiments and stored at 4 °C. For the assembly of NPCs, subunits were dispersed in 50 mM Tris-HCl, pH 7.5, 0.45 M NaCl, 1 mM DTT, 0.5 mM phenylmethylsulfonyl fluoride (PMSF), while genome and PSS were in 10 mM Tris-HCl, pH 7.5. The final concentration of subunits was 1 g L$^{-1}$ (24.6 μM) and the amount of genome was adjusted according to the desired subunit-to-genome mass ratio. An additional volume of 10 mM Tris-HCl, pH 7.5—with 30 mM CaCl$_2$ for the experiment with divalent cations—was used to adjust the ionic strength to 0.1 M. For the relaxation of NPCs into virions, pre-assembled NPCs were prepared in 10 mM Tris-HCl, pH 7.5, 1 mM DTT, 0.5 mM PMSF with a subunit concentration of 1.72 g L$^{-1}$ (42.3 μM), and they were mixed with an equal volume of 10 mM sodium acetate, pH 4.8, and 90 mM NaCl. The temperature of the experiments was 20 °C unless otherwise stated.

**Time-resolved small-angle X-ray scattering**. Scattering patterns were recorded at the ID02 beamline of the European Synchrotron Radiation Facility (Grenoble, France). The sample-to-detector distance was set to 3.0 m, which provided scattering wavenumbers $q$ ranging from $3.3 \times 10^{-3}$ to 0.26 Å$^{-1}$. The experimental setup included a stopped-flow apparatus (BioLogic SFM-400, France) equipped with four independent syringes. Beam exposure time was set to 5 ms. The two-dimensional scattering images were radially averaged and the uncertainty on the scattering intensity was computed as the standard deviation of the photon counts in the pixels within each bin. Intensities were converted into absolute units after subtraction of the contribution of buffer solutions. The forward scattering intensity and the radius of gyration were determined by using the DATRG tool of the ATSAS suite[50]. The ab initio reconstruction of NPCs was carried out with the program DAMMIN[50], which minimized the discrepancy between the calculated scattering pattern of a model of densely packed beads and the experimental pattern of NPCs smoothed by the program GNOM[50].

The forward scattering intensity $I_0 \equiv I(q = 0)$ of a NPC solution can be written as:

$$I_0 = \Delta b_S^2 X_S + \sum_{N=0}^{+\infty} X_N (\Delta b_G + \Delta b_S N)^2 = f(\langle N \rangle, \langle N^2 \rangle), \qquad (2)$$

with $\Delta b_G$ and $\Delta b_S$ the excess scattering lengths of genome and subunits respectively, $X_N$ the molar concentration of $N$th-order NPCs ($X_0 \equiv X_G$ the molar concentration of bare genome) and $X_S$ the molar concentration of free subunits. In reality, the limit to infinity is a convenient approximation and $N$ should be bounded by a maximal value (see Supplementary Methods for further discussion).

The mean number of subunits per NPC is defined as $\langle N \rangle = \sum_{N=0}^{+\infty} N X_N / \sum_{N=0}^{+\infty} X_N$.

Small-angle scattering is an ensemble-averaging technique and it is thereby not possible to infer the distribution of $N$ solely from measured intensities. In particular, we cannot calculate $\langle N \rangle$ from Eq. (2) without the knowledge of the standard deviation of $N$. However, we can find an upper limit $\langle N \rangle_{up}$, which verifies $\langle N \rangle_{up} \geq \langle N \rangle$ regardless the probability distribution of $N$ (see Supplementary Methods for justification):

$$\langle N \rangle_{up} \equiv \sqrt{\Gamma^2 + \frac{I_0 - I_0^*}{\Delta b_S^2 c_G}} - \Gamma, \qquad (3)$$

with $\Gamma = \Delta b_G / \Delta b_S - 1/2$ and $I_0^* \equiv \Delta b_G^2 c_G + \Delta b_S^2 c_S$ the forward scattering intensity produced by the mixture of subunits and genome prior to complexation. $c_G$ and $c_S$ are the total molar concentrations of genome and subunits, respectively.

**Small-angle neutron scattering**. SANS measurements were carried out at the PAXY beamline of the Laboratoire Léon Brillouin (Saclay, France). Two wavelengths of 6 and 8.5 Å, combined with sample-to-detector distances of 1, 3, and 5 m, were employed to cover a range of $q$ values from $4 \times 10^{-3}$ to 0.5 Å$^{-1}$. Samples were exposed for 1 h and the scattering intensities were recorded on a two-dimensional multi-wire detector. The scattering images were radially averaged and the uncertainty on the scattering intensity was computed as the standard deviation of the neutron counts in the pixels within each bin. Buffer solutions containing 68% D$_2$O were used to contrast match the genome.

**Cryo-transmission electron microscopy**. Three microliters of sample solution was deposited onto a holey carbon grid (Quantifoil R2/2) ionized by glow discharge. The grid was blotted with a filter paper and directly plunged into liquid ethane cooled down by liquid nitrogen using a FEI Vitrobot. The grids were stored in liquid nitrogen until use. The frozen samples were transferred into a Gatan 626 cryo-holder and observed at −180 °C via a JEOL JEM-2010 microscope equipped with a 200-kV field emission gun. The samples were imaged with a magnification of × 50,000 using a minimal dose system and the images were collected with a Gatan Ultrascan 4K CCD camera at 1.5 and 2 μm of defocus.

**Data availability**. The authors declare that the data supporting the findings of this study are available within the article and its Supplementary Information files or from the corresponding author on reasonable request.

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

## Acknowledgements

We thank Emmanuel Trizac for fruitful discussions on electrostatics. M.C. is supported by the "IDI 2016" project funded by the IDEX Paris-Saclay, ANR-11-IDEX-0003-02. D.L.-H. was partly funded by the Triangle de la Physique (contract 2012-065T). J.C. acknowledges a scholarship from the China Scholarship Council (CSC). G.T. acknowledges financial support from the Agence Nationale de la Recherche (contract ANR-16-CE30-0017-01). The electron microscopy imaging is supported by "Investissements d'Avenir" LabEx PALM (ANR-10-LABX-0039-PALM). We also acknowledge the European Synchrotron Radiation Facility (ESRF) and the Laboratoire Léon Brillouin (LLB) for allocation of synchrotron and neutron beam time on ID02 and PAXY beamlines, respectively. Molecular dynamics simulations have been performed at the "Centre de Calcul Scientifique en Région Centre" facility (CCRS-Orléans, France) under the CAS-CIMODOT program.

## Author contributions

M.C., D.L.-H., and G.T. designed the research. M.C., D.L.-H., and G.T. prepared the samples. M.C., D.L.-H., J.C., S.B., S.C., D.C., J.D., J.M., M.Z., and G.T. performed the experiments. M.C., D.L.-H., D.C., S.B., and G.T. analyzed and contributed to interpreting the data. G.T. wrote the paper.

## Additional information

**Competing interests:** The authors declare no competing interests.

