## [Peer Review File · Nature Communications]

Reviewers' comments:

Reviewer #1 (Remarks to the Author):

In this paper, the authors describe the results of direct observation of a virus (CCMV) assembly using TR-SAXS with a high brilliance synchrotron source. The technique used is excellent, and the data are of good quality. Although the authors discussed the results based on two simple models, nucleation-growth pathway and cooperative pathway, the experimental data looks not so simple for me. My questions and comments are listed below.

1) Gelbart and collaborators have shown that, during the protocapsid (neutral pH) stage of assembly, essentially all capsid proteins (CPs) are bound to RNA regardless of the CP/RNA ratio (ref. 8). If this is true, RNA maximally bound CPs, and the mean number of subunits (dimers) per nucleoprotein complex (NPC) should be close to 150 because the experiment was done under the charge matching conditions ($\rho=6$, that is, CP:RNA molar ratio=300:1). I could not understand the reason that $\langle N \rangle_{up} = 75$ was the final value attained in the assembly experiments at pH 7.5 (Figure 1b). The authors should describe their interpretation on this number.

2) In this context, I expected that the excessively bound CP was released during the virion formation at pH 5.2. Contrary to my expectation, $\langle N \rangle_{up}$ further increased during the virion formation (Figure 2a). Do the authors suggest that only 150 CPs are bound to RNA in the pre-assembled NPCs and that additional CP binding occurs during the virion formation? If so, it seems to be against the model of Gelbart's group (ref. 8). Please discuss about this discrepancy.

3) The authors used the double exponential functions for fitting of the decrease in R_g (Figure 1b), the increase in $\langle N \rangle_{up}$ (Figure 2a), the decrease in the scattering intensity (Figure 2c), and the decrease in R_g (Figure 3a). Although the authors assigned one of two phases to the binding (Figure 2a) or the structural rearrangement (Figures 1b, 2c, and 3a), they did not give any comments on the remaining phase. The authors should provide their interpretation for all observed phases.

4) As clearly seen in Figure 3a, $\langle N \rangle_{up}$ increased first and then decreased. This means that CPs bind excessively to PSS and then a part of bound CPs are released in the later phase. This kinetics should be fitted by a double exponential function at least.

Reviewer #2 (Remarks to the Author):

In a set of elegant time-resolved SAXS experiments, the authors examine how CCMV capsid protein assemble around RNA and polystyrene sulfonate (PSS) in vitro. They follow the assembly process across each phase of the reaction, and along the way, they use simple kinetic and statistical-thermodynamic models to interpret their measurements. For example, first-order rate equations determine three "relaxation" times: the time for CP to bind the RNA or PSS, the time for CP to unbind from RNA or PSS, and the time for RNA or PSS – dressed with CP – to shrink down to capsid size. The observations suggest different assembly pathways in different solution conditions. The nucleoprotein complexes (NPCs) are presumed to behave like micelles, and their equilibrium properties, such as the CP-RNA binding energy, the critical concentration of NPCs, and the average number of CPs bound to a NPCs, are analyzed from the statistical-thermodynamics of micelle self-assembly.

This work is one of the first to provide a time-resolved view of capsid assembly with sufficient resolution to make inferences about assembly pathways. Furthermore, it is to my knowledge the first report of different assembly pathways for the same capsid protein at different solution conditions. Thus, I consider it a highly significant advance for the field and certainly worthy of

being published in Nat. Comm. I have a few comments to improve the clarity of the manuscript though.

-The observation of both pathways is exciting, but a bit complicated by the fact that the assembly occurs in two steps. Based on work from the Knobler and Gelbart groups I gather that this is necessary for assembly around RNA in vitro, but could the authors comment on this? Did they try it one step assembly at lower pH? It would be good to at least mention this in the text.

-The discussion of the theoretical model used by the authors to analyze their assemblies should more clearly spell out its limitations. The authors briefly allude to the fact that they neglect repulsive interactions between adsorbed proteins in the discussion (and by the name of the model, although not all readers will be familiar with the term 'isodesmic'), but the consequences of this are not made clear. Most importantly, it seems that there must be attractive interactions between adsorbed proteins even at high pH, since they observe assembly at neutral pH around PSS. Similarly, in work from the Knobler-Gelbart group the experimental results supported weak but non-negligible interactions between capsid proteins at neutral pH. In addition, electrostatic repulsions between adsorbed proteins would be expected to play a role at high adsorption densities. Thus, the fit value of K can only be considered some effective sum of protein-RNA and protein-protein interactions averaged over the range of concentrations of proteins on the scaffold.

-In the derivation of the model (in the supplement), it would be helpful to note that you are assuming $c_s \gg c_g$ when obtaining the approximate relations for X_s (below eq. S3). Perhaps for clarity also note that the large N limit is assumed for the sums.

- Is it possible to experimentally obtain information about the distribution of the number of subunits bound on to genomes (rather than just the mean)? If possible this would provide a more stringent test of the model.

- The relaxation dynamics in low pH regime is relatively slow, relatively (e.g. Figure 2a). Is there a way to speculate whether capsid proteins are transferred between RNAs (i.e. via RNA-RNA collisions)?

-Have the authors tried assembly at lower ρ ? If yes, what happens?

-Last sentence of the first paragraph under the "Results" section: is there really enough data in the low- q regime of Fig. S2 to be confident of this fit? Can the authors provide an estimate of the error?

- Figures 1b and 1c: The text reads as if a solution prepared as in 1b undergoes a 2-fold dilution, but this must not exactly be the case since ρ has changed. Can the authors be more specific?

-Since concentrations are being expressed in molar units, there should be a factor of Avogadro's number in the expression for the diffusion limited reaction rate constant (below Eq. 1). Also, the phrase 'reaction rate' should be changed to 'reaction rate constant'.

-Similarly, the expression for K^{-1} below eq. S3 (in the supplement) should have a factor of Avogadro's number to be consistent.

-Page 11, last sentence: "upon binding of subunits, PSS readily collapses through a coil-globule transition...": Is this known, or surmised? If known, the authors should provide a reference, if surmised they should say so.

-Eq S3 is missing a factor X_S in the last term.

-Fig S3c: there are a lot of free capsid proteins, which seems different from the results described

in Cadena-Nava et al (J. Virol. 2012, 86(6):3318) and related articles from that group, where most of the protein is adsorbed to RNA. Can the authors comment on what is different in their system compared to the earlier work?

-page 7 (of the pdf) the authors attribute the "energy barrier" for protein adsorption to occlusion from other adsorbed proteins. However, this could also arise if there is an orientation dependence or a dependence on protein conformation for subunit adsorption (for instance, don't the authors conclude that the RNA binding domains are folded in solution?)

-Are the capsid objects stable in infinite dilution for either set of experiments (with RNA at low pH, PSS at high pH)?

p13: how does the result that assembly occurs in one step around PSS at neutral pH imply that the same occurs in vivo? There the proteins must be assembling around RNA, which the experiments don't see occurring at neutral pH.

Reviewer #3 (Remarks to the Author):

The manuscript describes results of in vitro assembly experiments on the packaging of a native viral RNA and a synthetic polyelectrolyte (PSS) by coat proteins of the cowpea chlorotic mottle virus CCMV. The impact of the role of acidity and ionic strength are investigated. The experimental study relies mostly on time-resolved small-angle X-ray scattering measurements and, to a much lesser extent, on cryo-EM imaging and MD simulations. Simple models are applied to extract characteristic times and (free) energies associated with the co-assembly processes.

Although I find both the findings and the topic of the study highly interesting, I cannot recommend the manuscript to be published in its current form. On the one hand, I am not convinced that the study is sufficiently comprehensive and definitive for a high-impact journal such as Nature Communications.

On the other hand, I take issue with the theoretical interpretations of the experiments. And, last but not least, it is not made clear why the manuscript should attract an audience outside the science community interested in the physical chemistry of viruses and virus-like particles outside of a biological context. This should be made clear if the authors decide to resubmit.

On a more technical side, there are a number of points that need addressing and/or clarification. Incidentally, it would have been rather useful to have page numbers in the manuscript to be able to refer to specific points in the main text. Also, many equations in the Supplementary Information are without number, which makes a referencing cumbersome.

1. I contest that the experiments prove a cooperative and nucleation-and-growth pathway for the co-assembly of CCMV and polyelectrolyte cargo, as is claimed in the manuscript. The authors only show that the co-assembly involves a minimum of two processes with separate time scales. One of these concerns the adsorption of the coat proteins onto the polynucleotides (or polyanions), and the other the rearrangement of the proteins into what seem like complete capsids, or what the authors call the amorphous nucleoprotein complexes. Three issues present themselves in this context.

First, that the co-assembly process involves these two processes has been suggested before, e.g., by Gelbart and collaborators. This calls into question the novelty of the claim. See also Perlmutter and Hagan, *Annu Rev Phys Chem.* 2015, vol 66, pp 217–239.

Second, cooperativity is not needed to produce two time scales in templated assembly. See, e.g., Cingil et al., *J Am Chem Soc* 2017, vol 139, pp 4962–4968. One just needs two processes characterized by sufficiently disparate time scales.

Third, what kind of cooperative process should be operative is not discussed. For instance, whether it is associated with conformational switching or the existence of a high free energy intermediate assemblies is not discussed let alone investigated. See, e.g., Hagan, *Adv Chem Phys* 2014, vol 155, pp 1 – 68.

2. The isodesmic assembly model used to analyse the assembly into nucleoprotein complexes is flawed. and four issues spring to mind immediately.

First, not the relevant expectation value of the protein aggregation number is calculated, but an upper limit. The reason is that the scattering experiments probe in effect a weight average, not a number average. Oddly enough, the model worked out in the Supplementary Information does allow one to calculate precisely that average, and it is not clear why this is not done.

Second, the first moment of the distribution is a very gradual function of the concentration of the various components, see the SI of the manuscript below eq. S2 in the middle of the page. In the SI and in the main text it is approximated by a micellization-like model that has all the hallmarks of cooperativity. The isodesmic model does in fact not exhibit a sharp critical concentration. As a result of this, the curve fitting of the experimental data shown, e.g., in Figure 1d, lead to an equilibrium constant that cannot be correct. Indeed, the mean degree of aggregation is not a function of the ratio of the genome-to-subunit mass ratio alone, but also of the concentration of one of the two components. This makes Figure 1d very peculiar - I expect the curve to be non-linear and the experimental data support this. The linear curve fit makes no sense.

Third, the isodesmic model cannot realistically describe adsorption on a finite template: how can an infinite number of proteins adsorb onto a finite template if this is driven by electrostatic attraction between template and proteins? I understand that this makes the mathematics a lot simpler, and maybe this is for this reason a useful approximation but one should at least mention and perhaps also discuss this.

Fourth, the dynamical isodesmic model is akin to the end-evaporation model of Cates and co-workers of living polymers. It has been analysed in great detail by Nyrkova and Semenov, *J Chem Phys* 2011, vol 134, no. 114902. Even this simple model does not give a simple exponential decay, and I expect the version on a template to also produce more complex relaxation dynamics. Hence, I do not understand why the authors do use a simple dynamical Langmuir adsorption model? This would give a neat single exponential relaxation.

3. Dilution experiments are very useful to probe the level of reversibility of the assembly. This is as far as I can tell only done for the coat proteins plus virus RNA at near-neutral pH. Why is this not repeated for the low pH and for the assembly experiments with PSS?

4. A systematic analysis of the cryo-EM images would be very useful. I mean by this a systematic evaluation of the size distribution, shape analysis and so on. We only get a taste for the sizes and shapes in Figures 1, 2 and 3.

5. In the last paragraph of the section the authors speculate why it is that PSS is able to close up the assemblies under conditions of near neutral pH even though native RNAs are not able to do that. The claim is that the PSS chains collapse through a coil-globule transition that allow the subunits to contact each other, and build up a closed shell. No proof is provided for this: this is pure speculation. If the closing up is caused by subunit-subunit interactions as the authors claim earlier on in the manuscript, then how these two claims can be made to be consistent with each other is wholly unclear to me.

6. In the Supporting Information, information on the structure factor of the viral RNA is provided. See Figure S2. The problem with ssRNA, however, is, is that it binds to other ssRNAs. Have the authors made sure that the concentration is low enough that complexation between RNAs does not take place? How has this been established? See also the work of Gopal and others.

7. In the Gromacs simulations of the coat protein, the RNA binding domains bind to negatively charged portions on the same structure, see Figure S8. This is not a real surprise, although one could ask the question how strongly this depends on the force fields used. The question also arises why these domains would bind to RNA, if they can also bind within the same protein.

Response to the reviewers

Reviewer 1

- 1. Gelbart and collaborators have shown that, during the protocapsid (neutral pH) stage of assembly, essentially all capsid proteins (CPs) are bound to RNA regardless of the CP/RNA ratio (ref. 8). If this is true, RNA maximally bound CPs, and the mean number of subunits (dimers) per nucleoprotein complex (NPC) should be close to 150 because the experiment was done under the charge matching conditions ($\rho=6$, that is, CP:RNA molar ratio=300:1). I could not understand the reason that $\langle N \rangle_{\text{up}} = 75$ was the final value attained in the assembly experiments at pH 7.5 (Figure 1b). The authors should describe their interpretation on this number.**

Garmann et al. [1-2] indeed reported that all the capsid subunits were bound to RNA regardless of the mass ratio, which would entail that the binding is irreversible, or in other words, that the binding energy of capsid subunits on the genome is high.

In contrast, it is one of our key findings that this binding energy is actually moderate ($\sim 7k_{\text{B}}T_0$ with k_{B} the Boltzmann constant and T_0 the room temperature). This result is consistent with the fact that the net charge of subunits is negative at pH 7.5, just like the genome, so that the attraction is only due to the small dipole moment of the subunits. It is also consistent with competition experiments [3] performed by the same team, where subunits initially bound to exogenous RNA eventually migrated toward native RNA; this process requires a free exchange of subunits between the two kinds of RNA, which would not be possible if the binding was irreversible. *Most importantly, a reversible binding entails that a significant fraction of the subunits remain free, which explains why we found only 75 subunits per NPC on average, in conditions where the subunit-to-genome molar ratio was 150 ($\rho = 6$).*

We have provided ample evidence supporting these findings: (i) our measurements performed independently by SANS and SAXS were in agreement; (ii) the dilution experiment (Fig. 1c) shows that the subunits can be released easily and thereby that the binding is reversible; and (iii) we observed a capture of subunits rather than a release during the relaxation of NPCs into virions (Fig. 3a), which clearly indicates that the NPCs contained less than 90 subunits on average.

Several sentences have been added in the Discussion (pp. 14-15) recalling the irreversible binding found by Garmann et al. and explaining why, in our case, the mean number of subunits per NPC was less than 150 at $\rho = 6$.

- 2. In this context, I expected that the excessively bound CP was released during the virion formation at pH 5.2. Contrary to my expectation, $\langle N \rangle_{\text{up}}$ further increased during the virion formation (Figure 2a). Do the authors suggest that only 150 CPs are bound to RNA in the pre-assembled NPCs and that additional CP binding occurs during the virion formation? If so, it seems to be against the model of Gelbart's group (ref. 8). Please discuss about this discrepancy.**

This comment is closely related to the previous one. Since our initial NPCs comprised 75 subunits, i.e., 150 capsid proteins, the completion of a virion required the capture of 15 free subunits. We observed indeed an increase of the scattering intensity during the formation of virions, which arose from the capture of free subunits (Fig. 3a). *If the subunits were initially in excess in NPCs, i.e., $\langle N \rangle_{\text{up}} > 90$, we would have observed a clear decrease of the scattering intensity due to their release, which was not the case. We also emphasize that, at the end of the relaxation, we did obtain well-formed virions, as shown by cryo-TEM in Fig. 3e.*

All our data are consistent with a weak binding energy of the subunits on the genome. Some of us (G.T. and M.C.) had a private discussion with Mauricio Comas-Garcia, Rees Garmann and William Gelbart. Their measurements were mostly carried out at 4 °C, so that we can expect a shift of the equilibrium in favor of a larger number of subunits bound on the genome. However, other effects of the temperature on the electrostatic interactions or on the protein and RNA conformations must be invoked to fully explain the irreversible binding that they observed.

- 3. The authors used the double exponential functions for fitting of the decrease in R_g (Figure 1b), the increase in $\langle N \rangle_{up}$ (Figure 2a), the decrease in the scattering intensity (Figure 2c), and the decrease in R_g (Figure 3a). Although the authors assigned one of two phases to the binding (Figure 2a) or the structural rearrangement (Figures 1b, 2c, and 3a), they did not give any comments on the remaining phase. The authors should provide their interpretation for all observed phases.**

Regarding the decrease of R_g (Fig. 1b and 4a), we think that the first phase may reflect a compaction of the genome and of the PSS due to the presence of subunits. Note that there is no reason why this phase should be as fast as the binding. The second phase may result from the partial association of bound subunits into capsid subassemblies.

For the relaxation dynamics of NPCs into virions, the two phases on Figs. 3a and 3c are of the same nature because the binding and the structural rearrangement are concomitant. The first phase may be fast because several binding sites in the NPCs are readily accessible to the free subunits, while the second phase takes more time because the dynamics becomes limited by the slow construction of the capsid.

Three sentences have been added in the Results (pp. 6, 10, 13).

- 4. As clearly seen in Figure 3a, $\langle N \rangle_{up}$ increased first and then decreased. This means that CPs bind excessively to PSS and then a part of bound CPs are released in the later phase. This kinetics should be fitted by a double exponential function at least.**

The small jump of $\langle N \rangle_{up}$ in Fig. 4a at $\sim 0.1-0.8$ s is not significant: first of all, its amplitude was small compared to the uncertainties. More importantly, in these kinetic experiments, subunits were rapidly mixed with PSS and injected into a measurement capillary within a few milliseconds. Air bubbles were often formed in the early time steps and gave rise to irrelevant data points which were removed (see Fig. R1a versus Fig. 4a). Air bubbles were easily recognized as they transiently induced a strong increase of the scattering intensity at low q -values (Fig. R1b). This was the most likely cause of the increase and decrease of $\langle N \rangle_{up}$ in the early time steps.

Figure R1 | Presence of air bubbles in the early time steps. **a.** $\langle N \rangle_{\text{up}}$ versus time after mixing subunits and 600-kDa PSS at pH 7.5 with a final ionic strength of 0.1 M. The mass ratio was $\rho = 8$. The shaded area indicates the jump of $\langle N \rangle_{\text{up}}$ and the red-circled data points were removed in the main text due to the presence of air bubbles. **b.** Two scattering patterns collected from the same experiment and showing the effect of air bubbles at low q -values (see the red pattern compared to the one in black).

Reviewer 2

1. **The observation of both pathways is exciting, but a bit complicated by the fact that the assembly occurs in two steps. Based on work from the Knobler and Gelbart groups I gather that this is necessary for assembly around RNA in vitro, but could the authors comment on this? Did they try it one step assembly at lower pH? It would be good to at least mention this in the text.**

We carried out one-step assembly experiments by mixing simultaneously subunits, genome and a buffer solution bringing the pH to 5.2 and the ionic strength to 0.1 M. In these conditions, both subunit-subunit and subunit-genome interactions were strong, and at the subunit concentrations required to obtain a reasonable scattering signal-to-noise ratio, i.e., >0.5 mg/mL, a lot of misassembled and aggregated objects were formed. As a result, we could not perform any satisfactory analysis on these data. Note that Garmann and coworkers [4] also obtained aggregates in similar conditions even though they proceeded by dialysis.

A sentence has been added in the Discussion (p. 15).

2. **The discussion of the theoretical model used by the authors to analyze their assemblies should more clearly spell out its limitations. The authors briefly allude to the fact that they neglect repulsive interactions between adsorbed proteins in the discussion (and by the name of the model, although not all readers will be familiar with the term ‘isodesmic’), but the consequences of this are not made clear. Most importantly, it seems that there must be attractive interactions between adsorbed proteins even at high pH, since they observe assembly at neutral pH around PSS. Similarly, in work from the Knobler-Gelbart group the experimental results supported weak but non-negligible interactions between capsid proteins at neutral pH. In addition, electrostatic repulsions between adsorbed proteins would be expected to play a role at high adsorption densities. Thus, the fit value of K can only be considered some effective sum of protein-RNA and protein-protein interactions averaged over the range of concentrations of proteins on the scaffold.**

Our isodesmic growth model (renamed templated assembly model) indeed averages all the interactions over N and the binding energy G_{bind} – or equivalently, the equilibrium constant K – comprises an effective contribution arising from the subunit-subunit interactions.

A sentence has been added in the Results (p. 8) mentioning the implicit contribution of the subunit-subunit interactions in the model parameters. Another sentence (p. 9) recalls that G_{bind} includes an effective contribution arising from these interactions as well. In response to the role of the electrostatic repulsion at high adsorption densities raised by the reviewer, we have specified, at the end of the same paragraph (p. 9), that the energy barrier to the binding of free subunits most likely stems from steric and electrostatic repulsions exerted by the bound subunits at high N .

- 3. In the derivation of the model (in the supplement), it would be helpful to note that you are assuming $c_s \gg c_g$ when obtaining the approximate relations for X_s (below eq. S3). Perhaps for clarity also note that the large N limit is assumed for the sums.**

The reviewer is right, we assumed $c_s \gg c_g$. It has been mentioned p. 2 of the Supplementary Information. We have also specified that N ran up to infinity (pp. 5, 8) and we have discussed the case where it has a maximal value in the Supplementary Information (pp. 3-4).

- 4. Is it possible to experimentally obtain information about the distribution of the number of subunits bound on to genomes (rather than just the mean)? If possible this would provide a more stringent test of the model.**

Small-angle scattering is an ensemble-averaging technique; it is therefore not possible to obtain information on the distribution of the number of subunits bound on the genome without assuming some model.

A sentence recalling this limitation has been added in the Methods (p. 18).

- 5. The relaxation dynamics in low pH regime is relatively slow, relatively (e.g. Figure 2a). Is there a way to speculate whether capsid proteins are transferred between RNAs (i.e. via RNA-RNA collisions)?**

Some subunits can presumably be exchanged between NPCs during their relaxation into virions. However, the most important feature is that there must be a reservoir of free subunits to capture. Thus, the slowness of the process does not arise from the collisions between NPCs but rather from the rearrangement of the bound subunits into capsids, which makes it increasingly costly in terms of energy to insert new subunits.

A sentence mentioning the possibility evoked by the reviewer has been added in the Results (p. 11).

- 6. Have the authors tried assembly at lower rho? If yes, what happens?**

Unfortunately, we did not have enough time to vary the subunit-to-genome mass ratio for the relaxation of NPCs due to the limited amount of synchrotron beamtime available, typically 3 days per year.

- 7. Last sentence of the first paragraph under the "Results" section: is there really enough data in the low-q regime of Fig. S2 to be confident of this fit? Can the authors provide an estimate of the error?**

The fit was performed with 8 data points, which is a fair number since we only have 2 fitting parameters. The uncertainties were large on the 3 first data points, which induced an uncertainty on R_g of 11 (s.e.m.) Å. Note that the typical uncertainty on R_g for the time-resolved experiments was 1 (s.e.m.) Å.

This uncertainty has been stated in the Results (p. 5) as well as in the Supplementary Fig. 4.

- 8. Figures 1b and 1c: The text reads as if a solution prepared as in 1b undergoes a 2-fold dilution, but this must not exactly be the case since ρ has changed. Can the authors be more specific?**

Correct. These NPCs were in fact pre-assembled for the experiment depicted in Fig. 3a. It turned out that a simple check on the dilution produced interesting features and we decided to put these results in Fig. 1.

We have been more specific in the Results (p. 6) by stating that the NPCs were pre-assembled at an ionic strength of 10 mM and $\rho = 5.5$, then they were rapidly diluted twofold. The word “pre-assembled” has been added in the caption of Fig. 1c.

- 9. Since concentrations are being expressed in molar units, there should be a factor of Avogadro’s number in the expression for the diffusion limited reaction rate constant (below Eq. 1). Also, the phrase ‘reaction rate’ should be changed to ‘reaction rate constant’.**

The Avogadro constant has been added into the expression of k^+ and the word “reaction rate” has been replaced by “reaction rate constant” both in the text and in the Supplementary Information.

- 10. Similarly, the expression for K^{-1} below eq. S3 (in the supplement) should have a factor of Avogadro’s number to be consistent.**

The inverse of Avogadro constant has been added into the expression of K^{-1} in the Supplementary Information (p. 3).

- 11. Page 11, last sentence: “upon binding of subunits, PSS readily collapses through a coil-globule transition...”: Is this known, or surmised? If known, the authors should provide a reference, if surmised they should say so.**

PSS is known to be insoluble in water when its charge fraction drops below 30% owing to its hydrophobic backbone [5]. Therefore, we speculate that the RNA-binding domains of the bound subunits might neutralise a large fraction of the negative charges carried by the PSS. As a result, the latter, which is very flexible, might collapse through a coil-globule transition that could allow the bound subunits to contact each other and to build up a closed shell.

The explanation (p. 13) has been rephrased in a similar manner, to make it clear that the proposed mechanism is speculative.

- 12. Eq S3 is missing a factor X_S in the last term.**

It has been corrected.

- 13. Fig S3c: there are a lot of free capsid proteins, which seems different from the results described in Cadena-Nava et al (J. Virol. 2012, 86(6):3318) and related articles from that group, where most of the protein is adsorbed to RNA. Can the authors comment on what is different in their system compared to the earlier work?**

This comment is closely related to the two first comments of the reviewer 1. In short, we found out that the binding of subunits on the genome is reversible and that the involved energy is moderate, which entails that there must be indeed a lot of free subunits remaining in the solution. For further details, please see our responses #1 and #2 to the reviewer 1, as well as the corresponding modifications made in the text.

- 14. page 7 (of the pdf) the authors attribute the “energy barrier” for protein adsorption to occlusion from other adsorbed proteins. However, this could also arise if there is an orientation dependence or a dependence on protein conformation for subunit adsorption**

(for instance, don't the authors conclude that the RNA binding domains are folded in solution?)

This is true, there must be also an orientation dependence since a subunit has a non-negligible dipole moment, and we cannot exclude a change of the subunit conformation upon binding, notably at the level of the RNA-binding domain.

These two points have been added in the Results (p. 9).

15. Are the capsid objects stable in infinite dilution for either set of experiments (with RNA at low pH, PSS at high pH)?

Virions and PSS-filled capsids are stable against dilution. This means that the subunits making up the capsid are no longer able to exchange with the bulk solution. This has been already observed by the Gelbart group as well as by ourselves during sample preparation (in [2] and [6]).

This detail has been mentioned in the Results (pp. 13-14).

16. p13: how does the result that assembly occurs in one step around PSS at neutral pH imply that the same occurs in vivo? There the proteins must be assembling around RNA, which the experiments don't see occurring at neutral pH.

The experiments suggest that a low pH is required to form capsids in the presence of genome. However, an acidic environment is very unlikely in the cytoplasm of plant cells. Interestingly, PSS-filled capsids are formed at neutral pH, which demonstrates that the subunits can make up a capsid even in physiological conditions. We think therefore that this is not such a factor as pH that permits to assemble virions *in vivo*. We would rather imagine a constant flux of subunits expressed by the cell machinery and gradually forming NPCs, and, at some point, an excess of subunits in the environment would allow the capsid to close up irreversibly. In short, a slow increase of the subunit concentration until excess might thus be the key factor. This is of course speculative

Reviewer 3

Although I find both the findings and the topic of the study highly interesting, I cannot recommend the manuscript to be published in its current form. On the one hand, I am not convinced that the study is sufficiently comprehensive and definitive for a high-impact journal such as Nature Communications.

The manuscript presents a set of kinetic experiments performed at *different ionic strengths, genome concentrations, pHs, temperatures, and involving the native genome as well as a synthetic polyelectrolyte*. Most of the experiments relied on TR-SAXS, which can only be carried out with a high brilliance synchrotron source, whose access is competitive and limited (a few days per year). Data analysis was performed with a minimum of assumptions, in particular $\langle N \rangle_{\text{up}}$ and R_g were chosen because they are model-independent. A simple model was used to get further information on the thermodynamics of subunit binding at neutral pH and involved a small number of fitting parameters. Finally, some of our results were cross-validated by SANS and cryo-TEM. We hope that all these elements will help convince better the reviewer.

On the other hand, I take issue with the theoretical interpretations of the experiments. And, last but not least, it is not made clear why the manuscript should attract an audience outside the science community interested in the physical chemistry of viruses and virus-like particles outside of a biological context. This should be made clear if the authors decide to resubmit.

Understanding the self-assembly mechanisms of viruses is of great importance for developing

therapeutic strategies inhibiting viral replication. But viral capsids are also a source of inspiration for biomimetic nanocages transporting heterologous nucleic acids, drugs, and nanoparticles in live imaging, and especially for synthetic delivery systems made of oppositely-charged components such as polymer- and lipid-DNA complexes for gene delivery applications [7]. CCMV is a minimalist and generic viral system, and it has been widely used in physical and nanotechnological studies. We expect therefore the dynamic pathways elucidated for CCMV to shed light on the mechanisms at work for a wide range of other viruses, viral nanocages and synthetic delivery systems. The end of the Discussion (p. 17) has been amended accordingly.

Incidentally, it would have been rather useful to have page numbers in the manuscript to be able to refer to specific points in the main text. Also, many equations in the Supplementary Information are without number, which makes a referencing cumbersome.

We have numbered the pages and the equations as requested.

- 1. I contest that the experiments prove a cooperative and nucleation-and-growth pathway for the co-assembly of CCMV and polyelectrolyte cargo, as is claimed in the manuscript. The authors only show that the co-assembly involves a minimum of two processes with separate time scales. One of these concerns the adsorption of the coat proteins onto the polynucleotides (or polyanions), and the other the rearrangement of the proteins into what seem like complete capsids, or what the authors call the amorphous nucleoprotein complexes. Three issues present themselves in this context. First, that the co-assembly process involves these two processes has been suggested before, e.g., by Gelbart and collaborators. This calls into question the novelty of the claim. See also Perlmutter and Hagan, *Annu Rev Phys Chem.* 2015, vol 66, pp 217–239.**

Gelbart and collaborators have investigated the equilibrium states of CCMV self-assembly. To our knowledge, they have never performed any time-resolved studies so that the kinetic pathways were discussed in hypothetical terms. *Here, we provide a comprehensive set of data on the subunit binding and structural relaxations at subsecond and nanometer resolution, which give a direct experimental evidence of both kinetic pathways.* The review of Perlmutter and Hagan [8] recalls the (equilibrium) experiments of the Gelbart group, as well as a TR-SAXS study on SV40 and numerical simulations that we duly cited in the Introduction (refs. [36,22-24]).

Second, cooperativity is not needed to produce two time scales in templated assembly. See, e.g., Cingil et al., *J Am Chem Soc* 2017, vol 139, pp 4962–4968. One just needs two processes characterized by sufficiently disparate time scales.

We understand from the remarks of the reviewer that the word “cooperative” could be misleading. The cooperativity did not occur during the binding of subunits onto the genome, as meant by the reviewer, but rather during the rearrangement, when both components were associated. We have now adopted the terminology of Hagan and coworkers, and we have renamed this pathway “en masse” [8]. *Since we observed that the subunits were binding onto the genome rapidly and the resulting complexes rearranged slowly, our data exactly verified the definition of the en masse pathway identified in numerical simulations.*

Third, what kind of cooperative process should be operative is not discussed. For instance, whether it is associated with conformational switching or the existence of a high free energy intermediate assemblies is not discussed let alone investigated. See, e.g., Hagan, *Adv Chem Phys* 2014, vol 155, pp 1 – 68.

The rearrangement into virions requires to overcome the energy barrier related to the electrostatic repulsion between subunits and to the conformational change of the subunits necessary for their arrangement onto an icosahedral lattice. The case of PSS-filled capsids is striking since it seems that the proximity of the subunits induced by PSS allows them to

overcome this barrier in conditions where they would not do so in the presence of genome. This has been clarified in response to the other reviewers in the Results (p. 13).

- 2. The isodesmic assembly model used to analyse the assembly into nucleoprotein complexes is flawed. and four issues spring to mind immediately. First, not the relevant expectation value of the protein aggregation number is calculated, but an upper limit. The reason is that the scattering experiments probe in effect a weight average, not a number average. Oddly enough, the model worked out in the Supplementary Information does allow one to calculate precisely that average, and it is not clear why this is not done.**

The self-assembly of capsids packaging genome is a complex process and there is currently no model accounting for the dynamics related to the binding of subunits and the rearrangement into virions. The purpose of the isodesmic growth model was to estimate some thermodynamic parameters (critical concentration and binding free energy) as well as the diffusion coefficient of the subunits on the genome during the NPC self-assembly only. It could not be used as such for the formation of virions because it cannot yield a purely monodisperse population of virions (along with free subunits) at the equilibrium, unless one questionably complexifies it and adds more unknown parameters. *We proposed an analytically tractable model with just a few measurable parameters.* All the other experiments were interpreted in terms of $\langle N \rangle_{\text{up}}$, which gives an upper limit for the mean number of subunits per NPC in the absence of information on the full distribution of N . $\langle N \rangle_{\text{up}}$ is model-independent and turns out to be exactly the mean number of subunits for a monodisperse population like the virions. The isodesmic growth model has been renamed into “templated assembly model” for a better understanding by the general audience. Its purpose has been clarified (p. 8) and the Results has been slightly rearranged to put the data related to the model at the end of the subsection (pp. 8-9). Accordingly, Figure 1 has been split and a new Figure 2 has been created. The binding free energy has been reassessed by using the templated assembly model for a better consistency and its value has barely changed ($-6.7k_{\text{B}}T_0$ versus $-6.9k_{\text{B}}T_0$ before by using $\langle N \rangle_{\text{up}}$).

Second, the first moment of the distribution is a very gradual function of the concentration of the various components, see the SI of the manuscript below eq. S2 in the middle of the page. In the SI and in the main text it is approximated by a micellization-like model that has all the hallmarks of cooperativity. The isodesmic model does in fact not exhibit a sharp critical concentration. As a result of this, the curve fitting of the experimental data shown, e.g., in Figure 1d, lead to an equilibrium constant that cannot be correct. Indeed, the mean degree of aggregation is not a function of the ratio of the genome-to-subunit mass ratio alone, but also of the concentration of one of the two components. This makes Figure 1d very peculiar - I expect the curve to be non-linear and the experimental data support this. The linear curve fit makes no sense.

Figure R2 depicts the solution of Eq. (S4). We can see a sharp critical concentration at $c_{\text{S}} \approx K^{-1}$ so that the approximation (S5) is well justified.

Figure R2 | Critical concentration at the equilibrium. Solution of Eqs. (S4) and (S6) with $c_G = 0.2 \mu\text{M}$ and $K^{-1} = 14 \mu\text{M}$, which are typical concentrations encountered in the experiments. The dashed line is an unstable branch.

In the experiments of NPC self-assembly, the subunit concentration c_S was maintained fixed at $24.6 \mu\text{M}$. According to Eq. (S7), we can write $\langle N \rangle(t \rightarrow +\infty) \propto (1 - K^{-1}/c_S)\rho$. Consequently, the linear fit of Fig. 2c does make sense and gives a correct estimate of K^{-1} . The above figure has been inserted in the Supplementary Information and the linear relationship has been made explicit in the Results (p. 8).

Third, the isodesmic model cannot realistically describe adsorption on a finite template: how can an infinite number of proteins adsorb onto a finite template if this is driven by electrostatic attraction between template and proteins? I understand that this makes the mathematics a lot simpler, and maybe this is for this reason a useful approximation but one should at least mention and perhaps also discuss this.

In fact, we can find a complex equation relating X_S to c_S after introducing the maximal number of subunits per NPC, N_{max} . Assuming $c_S \gg c_G$ as before, $\langle N \rangle(t \rightarrow +\infty)$ can be conveniently described by a piecewise function of c_S (Eq. (S11)). However, as seen on Fig. 2c, $\langle N \rangle(t \rightarrow +\infty)$ varied linearly with ρ , which means that the NPCs were far from being saturated by bound subunits. There was thereby no need to explicitly introduce a finite parameter N_{max} . The justification has been added in the Results (pp. 8-9) and the equations have been put in the Supplementary Information.

Fourth, the dynamical isodesmic model is akin to the end-evaporation model of Cates and co-workers of living polymers. It has been analysed in great detail by Nyrkova and Semenov, J Chem Phys 2011, vol 134, no. 114902. Even this simple model does not give a simple exponential decay, and I expect the version on a template to also produce more complex relaxation dynamics. Hence, I do not understand why the authors do use a simple dynamical Langmuir adsorption model? This would give a neat single exponential relaxation.

We guess that the reviewer meant “[...] I do not understand why the authors do not use a simple [...]”. The Langmuir adsorption model describes the adsorption of free molecules onto a surface. It assumes that the fraction of adsorbed molecules can be neglected so that the concentration of free molecules is constant. Consequently, the dynamics reduces to a simple exponential decay. In the NPC self-assembly experiment at $\rho = 8$, nearly 40% of the subunits were bound on the genome at the equilibrium (Supplementary Fig. 5c), so the Langmuir adsorption model is not realistic here. The templated assembly model is closer to the actual system (molecules binding on other molecules) and provides the time evolution of the distribution of N , which is necessary for fitting the forward scattering intensity.

- 3. Dilution experiments are very useful to probe the level of reversibility of the assembly. This is as far as I can tell only done for the coat proteins plus virus RNA at near-neutral pH. Why is this not repeated for the low pH and for the assembly experiments with PSS?**

This comment is similar to the comment #15 of the reviewer 2. Virions and PSS-filled capsids are stable against dilution. Once the capsid has closed up, the subunits are no longer able to exchange with the bulk solution, at least over the time frame of a TR-SAXS experiment.

- 4. A systematic analysis of the cryo-EM images would be very useful. I mean by this a systematic evaluation of the size distribution, shape analysis and so on. We only get a taste for the sizes and shapes in Figures 1, 2 and 3.**

The image analysis of NPCs is difficult because they are amorphous. We have however estimated an equivalent diameter of 27 ± 4 (s.d.) nm with an average aspect ratio of 1.13 (p. 8). The diameter of virions at pH 5.2 has been estimated to be 29 ± 1 (s.d.) nm (p. 11) and that of PSS-filled capsids at pH 7.5, 23 ± 1 (s.d.) nm, by excluding the few capsids that are around 29 nm (p. 13).

- 5. In the last paragraph of the section the authors speculate why it is that PSS is able to close up the assemblies under conditions of near neutral pH even though native RNAs are not able to do that. The claim is that the PSS chains collapse through a coil-globule transition that allow the subunits to contact each other, and build up a closed shell. No proof is provided for this: this is pure speculation. If the closing up is caused by subunit-subunit interactions as the authors claim earlier on in the manuscript, then how these two claims can be made to be consistent with each other is wholly unclear to me.**

This comment is similar to the comment #11 of the reviewer 2. It is an experimental fact that PSS-filled capsids can be formed at neutral pH while genome-filled capsids cannot. We attempted to explain this finding via the hydrophobicity and flexibility of PSS. When a large charge fraction of PSS is neutralized by the RNA-binding domains of subunits, PSS might collapse and thereby lower the energy barrier to the association of bound subunits. RNA is hydrophilic and stiff because base-paired, so that it cannot collapse in the same way.

- 6. In the Supporting Information, information on the structure factor of the viral RNA is provided. See Figure S2. The problem with ssRNA, however, is, is that it binds to other ssRNAs. Have the authors made sure that the concentration is low enough that complexation between RNAs does not take place? How has this been established? See also the work of Gopal and others.**

The genome concentration was below $0.5 \mu\text{M}$, which makes any complexation unlikely, especially for such long nucleic acids. Moreover, we measured a radius of gyration of 140 \AA , which is in agreement with cryo-TEM measurements performed by Garmann and coworkers giving a size of about 30 nm [1-2]. Also, the forward scattering intensity I_0 was consistent with a molecular weight around 900 kDa. At last, and perhaps more importantly, if several ssRNA were associated, we would not obtain $T = 3$ viral particles at low pH but larger particles or even multiplets (several capsids associated to share the same cargo) [2], which was not the case.

- 7. In the Gromacs simulations of the coat protein, the RNA binding domains bind to negatively charged portions on the same structure, see Figure S8. This is not a real surprise, although one could ask the question how strongly this depends on the force fields used. The question also arises why these domains would bind to RNA, if they can also bind within the same protein.**

The all-atom GROMACS simulations were performed with the AMBER force fields, which are well-established, proven force fields for molecular dynamic simulations of proteins. Even if the RNA-binding domain is folded along the body of the protein, some of its cationic charges are still exposed and the whole protein has a nonvanishing dipole moment. We think however that

its binding affinity for the genome is less than claimed before [3], hence the observed reversibility. The binding might also require a conformational switch of the RNA-binding domain that might give rise to an energy barrier accounting for the reduced diffusion coefficient calculated in the first subsection.

This point has also been discussed in our response to the comment #14 of the reviewer 2.

References

- [1] Garmann, R.F. et al. Role of electrostatics in the assembly pathway of a single-stranded RNA virus. *J. Virol.* **88**, 10472-10479 (2014).
- [2] Garmann, R.F., Comas-Garcia, M., Knobler, C.M. & Gelbart, W.M. Physical principles in the self-assembly of a simple spherical virus. *Acc. Chem. Res.* **49**, 48-55 (2016).
- [3] Comas-Garcia, M., Cadena-Nava, R.D., Rao, A.L.N., Knobler, C.M. & Gelbart, W.M. In vitro quantification of the relative packaging efficiencies of single-stranded RNA molecules by viral capsid protein. *J. Virol.* **86**, 12271-12282 (2012).
- [4] Garmann, R.F., Comas-Garcia, M., Gopal, A., Knobler, C.M. & Gelbart, W.M. The assembly pathway of an icosahedral single-stranded RNA virus depends on the strength of inter-subunit attractions. *J. Mol. Biol.* **426**, 1050-1060 (2014).
- [5] Baigl, D., Seery, T.A.P. & Williams, C.E. Preparation and characterization of hydrosoluble, partially charged poly(styrenesulfonate)s of various controlled charge fractions and chain lengths. *Macromolecules* **35**, 2318-2326 (2002).
- [6] Tresset, G. et al. Weighing polyelectrolytes packaged in viruslike particles. *Phys. Rev. Lett.* **113**, 128305 (2014).
- [7] de Iliarduya, C.T., Sun, Y. & Duezguenes, N. Gene delivery by lipoplexes and polyplexes. *Eur. J. Pharm. Sci.* **40**, 159-170 (2010).
- [8] Perlmutter, J.D. & Hagan, M.F. Mechanisms of virus assembly. *Annu. Rev. Phys. Chem.* **66**, 217-239 (2015).

Reviewers' comments:

Reviewer #1 (Remarks to the Author):

Responding to my previous comments #1 and #2, the authors have mentioned the difference between their observation and the results reported by Gelbart's group. They have also mentioned the discussion with Gelbart's group and alluded to a possibility that the different observations arose from the difference in the experimental temperature. In addition to this difference, I noticed the fact that the authors used native (CCMV) RNA whereas Gelbart's group used brome mosaic virus RNA. I recommend that the authors call readers attention to these differences in the text.

To my previous comment #3 that the authors should provide their interpretation for all observed phases, the authors have added short sentences about their interpretation for kinetic phases. However, those are not convincing and too speculative. More specifically, is it necessary to fit the relaxation kinetics shown in Figure 3a and in Supplementary Figure 8 with double exponential functions? If it is meaningful, why did the authors show the Arrhenius plot (Figure 3d) only for the slower phase? Why did not the authors show the time dependent intensity change at $q=0.033 \text{ \AA}^{-1}$ (Figure 3c) for other temperatures? Whether are there CP concentration dependence for all kinetic phases observed in Figures 1, 3 and 4?

In relevant to these questions, I have a comment to the statement (page 11, line 5) that since $\tau_{\text{bind}} \approx \tau_{\text{struc}}$, the system likely proceeded through a nucleation-growth mechanism where the capsid was building up by capturing free subunits while packaging the genome. This is not correct. If the reaction proceeds as

$\text{genome} + n\text{CP} \rightleftharpoons \text{NPC} \rightarrow \text{virion}$,

and if the binding equilibrium of CP and RNA is rapid, the rate determining step of the reaction is conversion from NPC to virion. In such a case, the number of bound CP increases with the same rate as the conversion from NPC to virion. In this case, however, NPC is a stable (observable) intermediate but not a nucleus. I think, therefore, we should not use the term "nucleation-growth mechanism" in this case. The term "nucleation-growth mechanism" should be used for phenomena with all-or-none property such as the helix-coil transition, where the nucleus is too unstable to be detected.

In the response to my previous comment #4, the authors have stated that the small jump of $\langle N \rangle_{\text{up}}$ in Fig. 4a at $\sim 0.1-0.8 \text{ s}$ is not significant because of its small amplitude. Furthermore, they suggested the possibility of artefact from stopped-flow mixing. I know that the artefact from air bubbles is frequently observed in the stopped-flow experiments. Usually, therefore, several shots (stopped-flow mixings and observations) are repeated, and the experimentalist address whether the observation is artefact or not.

If the authors repeated such experiments, please mention whether another data did not show the increase in the time range $0.1-0.8 \text{ s}$.

Reviewer #2 (Remarks to the Author):

In my opinion the authors have responded satisfactorily to the reviewer comments and I recommend publication.

I have just one request for clarification:

In figure 1c, why doesn't the R_g of RNA increase when it loses CPs to solution; i.e., why don't we see the opposite of the trend described as a structural transition upon protein binding in Fig. 1a? Is

this because of different solution conditions between the final state in 1a and initial state in 1c? It would be helpful to elaborate on this. Also, for Fig. 3a the text states "The radius of gyration did not change significantly during this time frame" but R_g seems to rise systematically over this time frame. Explanation would be helpful.

Reviewer #3 (Remarks to the Author):

The revisions have improved the manuscript very significantly, and I am content with the response of the authors to the referee reports. The manuscript is a pleasure to read, it has a clear message and structure, and it is now clear to me why the work is of importance to a wider scientific audience. In my view the manuscript can be published as is. Congratulations to the authors.

There remain a few minor points that the authors may wish to consider, but I am not insisting they should.

1. I agree that their estimate for $\langle N \rangle_{\text{up}}$ is model-free and useful. Yet, I fail to see what the problem is trying to establish $\langle N^2 \rangle$ to get a better estimate for $\langle N \rangle$. It can be calculated directly from the templated assembly model the authors put forward in the SI. On a more model-free note, there is a fundamental expression for the variance σ_N^2 of the assemblies that the authors seem not familiar with: $\sigma_N^2 = \partial \langle N \rangle / \partial \ln c_S$, with c_S the free subunit concentration. See early works of Isrealachvili on micelle thermodynamics. It maps straightforwardly to the templated assembly case. One could use this to construct a differential equation in terms of $\langle N \rangle$ in equation S14.

2. In response to one of the comments on my remarks relating to applying a dynamical Langmuir theory to the problem in hand, it does not need to fix the concentration of subunits in the bulk solution - the problem can be set up self-consistently. If one does, the equations are non-linear and the solution not a simple exponential decay. They do become approximately linear in the limit of excess protein.

3. In relation to the explanation why the linear synthetic polymer does get encapsulated at neutral pH and RNA, I would like to point at a recent work of the Zandi group that addresses this question, see arXiv:1801.02653. An earlier work from Zandi in 2013, cited in this manuscript, also addresses this issue.

4. I disagree with the statement that even if the net charge of the proteins is negative at pH = 7.5, the binding of a polyanion must be due to the small (?) dipole moment. It is well established that block copolymers consisting of positively and negatively charged blocks are able to bind charged homopolymers, even if the net charge of the block copolymers is zero or has the same sign as the homopolymer. This is due to screening of the interactions that mask the presence of the same-charge moieties if they are sufficiently removed from each other in space.

5. The authors in my view should add half a sentence around eq (2) that taking the limit to infinity is an approximation but not a serious one, and refer to the SI.

6. I think that the subsection title "NPCs relax into virions via a nucleation-growth pathway" is too strong and that the word "putative" should be added to indicate that it is a likely mechanism but that there is no definitive proof of this. See my earlier comments.

Second response to the reviewers

Reviewer 1

1. Responding to my previous comments #1 and #2, the authors have mentioned the difference between their observation and the results reported by Gelbart's group. They have also mentioned the discussion with Gelbart's group and alluded to a possibility that the different observations arose from the difference in the experimental temperature. In addition to this difference, I noticed the fact that the authors used native (CCMV) RNA whereas Gelbart's group used brome mosaic virus RNA. I recommend that the authors call readers attention to these differences in the text.

Garmann and coworkers [1,2] indeed used exogenous RNA derived from the genome of brome mosaic virus and of tobacco mosaic virus. It has been explicitly mentioned at the beginning of the Discussion (p. 14).

2. To my previous comment #3 that the authors should provide their interpretation for all observed phases, the authors have added short sentences about their interpretation for kinetic phases. However, those are not convincing and too speculative. More specifically, is it necessary to fit the relaxation kinetics shown in Figure 3a and in Supplementary Figure 8 with double exponential functions?

Fig. R1 shows that a double exponential decay function fits the experimental data depicted on Fig. 3a more reliably than a single exponential decay function. We have therefore also used the double exponential decay function for the other data depicted in Supplementary Fig. 9.

Figure R1 | Comparison between single and double exponential decay functions. top, $\langle N \rangle_{\text{up}}$ versus time as shown on Fig. 3a with a single (blue line) and a double (red line) exponential decay function. **bottom,** The corresponding residuals.

3. **If it is meaningful, why did the authors show the Arrhenius plot (Figure 3d) only for the slower phase?**

Since we looked for an activation energy limiting the relaxation of NPCs into virions, it is natural to consider the longest timescale because it is related to the highest energy barrier. Fig. R2 depicts an Arrhenius plot constructed with the shortest binding times of the relaxation experiments shown in Supplementary Fig. 9. The trend is not conclusive, which suggests that the fast phase might involve non-reaction-limited processes.

Figure R2 | Arrhenius plot for the fast phase. Logarithm of the shortest binding time τ_{bind}^s as a function of the inverse of temperature for the data shown in Supplementary Fig. 9.

4. **Why did not the authors show the time dependent intensity change at $q=0.033 \text{ \AA}^{-1}$ (Figure 3c) for other temperatures?**

The variations of the form factors of NPCs at $q = 0.033 \text{ \AA}^{-1}$ and at different temperatures has been added in Supplementary Fig. 10. The structural time τ_{struc} remained of the same order of magnitude as the binding time τ_{bind} (see Supplementary Fig. 9) for all the temperatures, which further supports the nucleation-growth pathway (now renamed “synchronous pathway”, see below).

A sentence has been added in the Results (p. 11).

5. **Whether are there CP concentration dependence for all kinetic phases observed in Figures 1, 3 and 4?**

The subunit concentration was kept constant in most of the experiments, but we varied the genome concentration for the self-assembly dynamics of NPCs (Fig. 1). We have therefore computed the binding time (τ_{bind}) by fitting $\langle N \rangle_{\text{up}}$ with a single exponential decay function and the two structural times (τ_{struc} and τ_{struc}^s) by fitting R_g with a double exponential decay function, for various values of ρ . Quite interestingly, while τ_{bind} did not change much, the structural times were both reduced at small mass ratios (Fig. R3). It can be explained by the fact that the NPCs

contained less subunits (e.g. 19 at $\rho = 2.45$) and thus, the degree of self-organization became weak and subsequently fast.

Figure R3 | Characteristic timescales for the self-assembly dynamics of NPCs at various ρ . Binding time τ_{bind} , structural time τ_{struc} and shortest structural time $\tau_{\text{struc}}^{\text{s}}$ as a function of the mass ratio ρ in a buffer solution at pH 7.5 with a final ionic strength of 0.1 M. Error bars are defined as s.e.m.

We did not have enough beamtime to investigate the concentration dependence on the NPC relaxation into virions (Fig. 3). But we performed an additional experiment for the self-assembly of PSS-filled capsids at $\rho = 6$ (see Fig. R4). However, after careful analysis, it turned out that the fraction of 'large' species was significant and the data could not be compared with those presented on Fig. 4 at $\rho = 8$.

Supplementary Fig. 5 has been added as well as a sentence in the Results (p. 6).

6. In relevant to these questions, I have a comment to the statement (page 11, line 5) that since $\tau_{\text{bind}} \approx \tau_{\text{struc}}$, the system likely proceeded through a nucleation-growth mechanism where the capsid was building up by capturing free subunits while packaging the genome. This is not correct. If the reaction proceeds as

and if the binding equilibrium of CP and RNA is rapid, the rate determining step of the reaction is conversion from NPC to virion. In such a case, the number of bound CP increases with the same rate as the conversion from NPC to virion. In this case, however, NPC is a stable (observable) intermediate but not a nucleus. I think, therefore, we should not use the term "nucleation-growth mechanism" in this case. The term "nucleation-growth mechanism" should be used for phenomena with all-or-none property such as the helix-coil transition, where the nucleus is too unstable to be detected.

We do not follow the reasoning of the reviewer: if the binding equilibrium of CP and RNA is rapid, then necessarily τ_{bind} is small, and τ_{struc} is given by the slow conversion from NPC to virion. Unlike what is stated by the reviewer, we arrive at $\tau_{\text{bind}} \ll \tau_{\text{struc}}$, which means an en masse pathway as expected.

We adopted the expression “nucleation-growth pathway” since it becomes well admitted in the field and several authors have been using it because the pathway is reminiscent of the assembly of empty capsids. However, since the reviewers 1 and 3 are somewhat opposed to it (see the comment #6 of the reviewer 3), we have used instead the expression “synchronous pathway” in order to simply qualify that the binding of subunits and the packaging of genome occur in a synchronous manner.

The words “nucleation-growth” have been replaced throughout the text and the Introduction has been slightly modified (p. 3) to relate the new expression “synchronous pathway” to previous works.

- 7. In the response to my previous comment #4, the authors have stated that the small jump of $\langle N \rangle_{\text{up}}$ in Fig. 4a at $\sim 0.1-0.8$ s is not significant because of its small amplitude. Furthermore, they suggested the possibility of artefact from stopped-flow mixing. I know that the artefact from air bubbles is frequently observed in the stopped-flow experiments. Usually, therefore, several shots (stopped-flow mixings and observations) are repeated, and the experimentalist address whether the observation is artefact or not. If the authors repeated such experiments, please mention whether another data did not show the increase in the time range 0.1-0.8 s.**

We did not repeat the experiment at $\rho = 8$ but we did another one at $\rho = 6$. As depicted by Fig. R4, we did not observe at $\rho = 6$ any increase of intensity in the time range 0.1-0.8 s in contrast with the case at $\rho = 8$. This comparison supports our previous statement, namely, the small jump of $\langle N \rangle_{\text{up}}$ is not significant and was most likely a remnant of artifacts arising from air bubbles.

The reviewer might wonder why we decided to show the data at $\rho = 8$ but not those at $\rho = 6$. Even though the latter have no artifacts, the fraction of 'large' species became significant, which rendered the determination of $\langle N \rangle_{\text{up}}$ unreliable.

Figure R4 | Self-assembly dynamics of PSS-filled capsids at $\rho = 8$ and 6. Forward scattering intensities I_0 versus time after mixing subunits and 600-kDa PSS at pH 7.5 with a final ionic strength of 0.1 M. The mass ratio was 8 (top) and 6 (bottom). The shaded area indicates the time range 0.1-0.8 s.

Reviewer 2

- In figure 1c, why doesn't the R_g of RNA increase when it loses CPs to solution; i.e., why don't we see the opposite of the trend described as a structural transition upon protein binding in Fig. 1a? Is this because of different solution conditions between the final state in 1a and initial state in 1c? It would be helpful to elaborate on this.**

We think that it was due to two competing effects: by releasing subunits, the NPCs tended to be less compact, but at the same time, these subunits did not contribute any longer to the radius of gyration of NPCs. Since only 17 subunits per NPC were released in that experiment, the effects cancelled out and R_g remained roughly unchanged.

A sentence has been added in the Results (pp. 6-7).

- Also, for Fig. 3a the text states “The radius of gyration did not change significantly during this time frame” but R_g seems to rise systematically over this time frame. Explanation would be helpful.**

Small-angle scattering is a low resolution technique. Consequently, we deemed that a variation of a few angstroms compared to 11 nm on a radius of gyration was not significant enough to draw a clear conclusion. Notice, however, that the timescale of this variation was close to τ_{bind} and would anyway support our claim, i.e., NPCs relax into virions via a synchronous pathway.

A sentence clarifying our previous statement has been added in the Results (p. 10).

Reviewer 3

We thank the referee for acknowledging the quality of the present version of our paper and that it could be published as is. Still, we have carefully considered the referee's comments and amended the work as follows.

- 1. I agree that their estimate for $\langle N \rangle_{\text{up}}$ is model-free and useful. Yet, I fail to see what the problem is trying to establish $\langle N^2 \rangle$ to get a better estimate for $\langle N \rangle$. It can be calculated directly from the templated assembly model the authors put forward in the SI. On a more model-free note, there is a fundamental expression for the variance σ_N^2 of the assemblies that the authors seem not familiar with: $\sigma_N^2 = \partial \langle N \rangle / \partial \ln c_S$, with c_S the free subunit concentration. See early works of Isrealachvili on micelle thermodynamics. It maps straightforwardly to the templated assembly case. One could use this construct a differential equation in terms of $\langle N \rangle$ in equation S14.**

The analytical expressions proposed by the reviewer are valid at the equilibrium only. They are also not appropriate for the formation of virions because they cannot account for a purely monodisperse population of virions at the equilibrium unless one complexifies the model. That is why we introduced $\langle N \rangle_{\text{up}}$, which is model-independent, valid out of equilibrium, and exact for a monodisperse population like the virions.

The values of $\langle N \rangle(t \rightarrow +\infty)$ shown in Fig. 2c were computed by solving Eq. (S14) consistently with the analytical expressions given by the templated assembly model at the equilibrium. In this particular case, we proceeded as suggested by the reviewer. Some clarifications on the procedure have been provided in the Supplementary Information (pp. 5-6).

- 2. In reponse to one of the comments on my remarks relating to applying a dynamical Langmuir theory to the problem in hand, it does not need to fix the concentration of subunits in the bulk solution - the problem can be set up self-consistently. If one does, the equations are non-linear and the solution not a simple exponential decay. They do become approximately linear in the limit of excess protein.**

We agree that a dynamical Langmuir model can be solved self-consistently but the advantage of simplicity is lost. We think that the templated assembly model is more comprehensive and closer to the actual system.

- 3. In relation to the explanation why the linear synthetic polymer does get encapsulated at neutral pH and RNA, I would like to point at a recent work of the Zandi group that addresses this question, see arXiv:1801.02653. An ealier work from Zandi in 2013, cited in this manuscript, also addresses this issue.**

The reference pointed out by the reviewer is quite relevant with our experiments on PSS-filled capsids. It demonstrates that flexible, linear chains require less free energy than stiff, branched ones and exhibit thereby a higher packaging efficiency.

The reference has been included and a sentence has been added in the Results (pp. 13-14).

- 4. I disagree with the statement that even if the net charge of the proteins is negative at pH =**

7.5, the binding of a polyanion must be due to the small (?) dipole moment. It is well established that block copolymers consisting of positively and negatively charged blocks are able to bind charged homopolymers, even if the net charge of the block copolymers is zero or has the same sign as the homopolymer. This is due to screening of the interactions that mask the presence of the same-charge moieties if they are sufficiently removed from each other in space.

As a matter of fact, we fully agree with the reviewer and we may not have responded clearly to the comment #1 of the reviewer 1. We did not mean that the subunits can bind the genome because the dipole moment is small, but because it is finite. As mentioned by the reviewer, block copolymers can bind charged homopolymers, even if the net charge has the same sign, provided that the same-charge moieties are sufficiently far apart, i.e., provided that the dipole moment, or higher-order moments, is large enough.

In the Discussion (p. 15), we pointed out that the dipole moment of subunits is reduced subsequently to the folding of the RNA-binding domains against the body of the proteins as seen in molecular dynamics simulations. We think that it can explain why the subunit-genome binding free energy is moderate in our experiments.

- 5. The authors in my view should add half a sentence around eq (2) that taking the limit to infinity is an approximation but not a serious one, and refer to the SI.**

A sentence has been added in the Methods (p. 18).

- 6. I think that the subsection title "NPCs relax into virions via a nucleation-growth pathway" is too strong and that the word "putative" should be added to indicate that it is a likely mechanism but that there is no definitive proof it this. See my earlier comments.**

This comment is related to the comment #6 of the reviewer 1. We have replaced the expression "nucleation-growth pathway" by "synchronous pathway" in order to simply qualify that the binding of subunits and the packaging of genome occur in a synchronous manner, which is precisely reflected by the condition $\tau_{\text{bind}} \approx \tau_{\text{struc}}$.

References

- [1] Garmann, R.F. et al. Role of electrostatics in the assembly pathway of a single-stranded RNA virus. *J. Virol.* **88**, 10472-10479 (2014).
- [2] Garmann, R.F., Comas-Garcia, M., Knobler, C.M. & Gelbart, W.M. Physical principles in the self-assembly of a simple spherical virus. *Acc. Chem. Res.* **49**, 48-55 (2016).

Reviewers' comments:

Reviewer #1 (Remarks to the Author):

I appreciate the authors providing the important information such as Figures R1-R4. This information helps me to understand what happens. Since this information is also useful for readers, it is good that Figure R3 has been presented as Supplementary Figure 5. Figures R2 should be also presented in the paper (at least as supplementary information). Related comments are shown below.

1) For NPC assembly, data shown in Figure R3 is quite interesting, as the authors have written. To my understanding, this data shows that the structural change of NPC is slower for the NPC containing the larger number of CPs. The compaction of genome, which is thought to correspond to the fast phase, may slow down if the larger number of CPs bind to RNA. The capsid subassembly, which is assigned to the slow phase, may be decelerate or accelerate depending on the mechanism. This point should be discussed.

2) For relaxation of NPCs into virions, the authors have provided the rate constants of the fast binding phase estimated from the increase in $\langle N \rangle_{up}$ (Figure R2). Although they have stated that the trend is not conclusive, which suggests that the fast phase might involve non-reaction-limited processes, I suggest omitting the data at 30 °C. As seen in Supplementary Figure 9, the curve fitting to the experimental data at 30 °C is clearly worse than those at other temperatures. This treatment seems to give a straight line of the activation energy of 20~30 kJ/mol.

3) I think the authors should not ignore the fast phase in the relaxation process of NPCs into virions (Figure 3 and Supplementary Figure 9). It seems that the kinetic amplitudes (the increments of bound CPs) of the fast and slow phases are comparable at 10 and 20 °C. At 40 °C, the kinetic amplitude of the fast phase is much larger than the slower phase. Furthermore, the reliability of the estimated rate constant is higher for the fast phase because there are many data points in the time range corresponding to the fast phase. Contrarily, data were acquired only for the time range shorter than the relaxation time of the slow phase at 10 and 20 °C.

4) Are the rate constants of the fast phases estimated from the $P(q=0.033)$ consistent with those from $\langle N \rangle_{up}$?

Reviewer #2 (Remarks to the Author):

In my view the authors have responded sufficiently to the reviewer comments, and I recommend publishing as is.

Third response to the reviewers

Reviewer 1

- 1. For NPC assembly, data shown in Figure R3 is quite interesting, as the authors have written. To my understanding, this data shows that the structural change of NPC is slower for the NPC containing the larger number of CPs. The compaction of genome, which is thought to correspond to the fast phase, may slow down if the larger number of CPs bind to RNA. The capsid subassembly, which is assigned to the slow phase, may be decelerate or accelerate depending on the mechanism. This point should be discussed.**

We agree with the mechanism proposed by the reviewer. A short discussion describing this interpretation has been added in the Results (p. 6).

- 2. For relaxation of NPCs into virions, the authors have provided the rate constants of the fast binding phase estimated from the increase in $\langle N \rangle_{up}$ (Figure R2). Although they have stated that the trend is not conclusive, which suggests that the fast phase might involve non-reaction-limited processes, I suggest omitting the data at 30 °C. As seen in Supplementary Figure 9, the curve fitting to the experimental data at 30 °C is clearly worse than those at other temperatures. This treatment seems to give a straight line of the activation energy of 20~30 kJ/mol.**

Following the reviewer' suggestion, we have presented Fig. R2 as Supplementary Fig. 11a. A linear fit performed by omitting the data point at 30°C gives an activation energy of 28.6 ± 15 kJ.mol⁻¹ as anticipated by the reviewer. A sentence has been added in the Results (p. 12).

- 3. I think the authors should not ignore the fast phase in the relaxation process of NPCs into virions (Figure 3 and Supplementary Figure 9). It seems that the kinetic amplitudes (the increments of bound CPs) of the fast and slow phases are comparable at 10 and 20 °C. At 40 °C, the kinetic amplitude of the fast phase is much larger than the slower phase. Furthermore, the reliability of the estimated rate constant is higher for the fast phase because there are many data points in the time range corresponding to the fast phase. Contrarily, data were acquired only for the time range shorter than the relaxation time of the slow phase at 10 and 20 °C.**

The kinetic amplitudes of the fast and slow phases at 10, 20 and 30°C are comparable (5~7 for the fast phase versus 8~10 for the slow one) while at 40°C the kinetic amplitude of the fast phase (~10) is larger than that of the slow phase (~4). However, the role of the kinetic amplitudes is beyond the scope of this article. As stated in our previous response, we focused on the longest timescale because it is related to the highest energy barrier. This barrier limits the relaxation process and thereby plays the major role in the dynamics.

We recognize nevertheless that the shortest timescale has interesting features. Accordingly, we have added Supplementary Fig. 11 and a few words introducing it have been inserted in the Results (p. 12).

- 4. Are the rate constants of the fast phases estimated from the $P(q=0.033)$ consistent with those from $\langle N \rangle_{up}$?**

Fig. R1 below shows that the rate constants of the fast phase, i.e., τ_{bind}^s and τ_{struc}^s , keep the same

order of magnitude across experiments performed at 10, 20, 30 and 40°C. This result supports further a synchronous pathway for the relaxation of NPCs into virions. Fig. R1 has been added as Supplementary Fig. 11b and has been introduced in the Results (p. 12).

Figure R1 | Short timescales for the relaxation of NPCs into virions. τ_{struc}^s versus τ_{bind}^s for experiments performed at 10, 20, 30 and 40°C. The timescales were obtained from the shortest decay time of the double exponential fits shown on Supplementary Figs. 9 and 10, i.e., from $\langle N \rangle_{\text{up}}$ and $P(q = 0.033 \text{ \AA}^{-1})$.

REVIEWERS' COMMENTS:

Reviewer #1 (Remarks to the Author):

The authors have responded sufficiently to my comments, and all experimental results are now accessible to readers. I recommend publication.